# EFFICIENT MULTI-SOURCE KNOWLEDGE TRANSFER BY MODEL MERGING

## ABSTRACT

While transfer learning is an advantageous strategy, it often overlooks the opportunity to leverage knowledge from numerous available models online. Addressing this multi-source transfer learning problem is a promising path to boost adaptability and cut re-training costs. However, existing approaches are inherently coarse-grained, lacking the necessary precision for granular knowledge extraction and the aggregation efficiency required to fuse knowledge from either a large number of source models or those with high parameter counts. We address these limitations by leveraging Singular Value Decomposition (SVD) to first decompose each source model into its elementary, rank-one components. A subsequent aggregation stage then selects only the most salient components from all sources, thereby overcoming the previous efficiency and precision limitations. To best preserve and leverage the synthesized knowledge base, our method adapts to the target task by fine-tuning only the principal singular values of the merged matrix. In essence, this process only recalibrates the importance of top SVD components. The proposed framework allows for efficient transfer learning, is robust to perturbations both at the input level and in the parameter space (e.g., noisy or pruned sources), and scales well computationally. Our code is provided in the supplementary.

## 1 INTRODUCTION

The increasing complexity of models and the immense computational costs associated with their training necessitate the efficient utilization of existing resources. Transfer learning Zhuang et al. (2020), which involves initializing networks with weights from a pretrained model, has emerged as a standard practice. This practice relies on foundational models, such as large-scale vision transformers Awais et al. (2025) and self-supervised models Caron et al. (2021), which learn robust and generalized representations from vast, general-purpose datasets (e.g., ImageNet, LAION-5B). By effectively leveraging this broad pre-existing knowledge, transfer learning significantly reduces the demand for extensive task-specific data, accelerates training, and enhances overall model performance across a wide range of computer vision tasks.

However, the wealth of specialized knowledge residing in other fine-tuned models remains largely untapped. Each model represents a valuable knowledge asset, with hundreds of thousands of versions publicly available on platforms like Hugging Face. Each new adaptation typically requires training from its original, pre-trained state, neglecting the specialized knowledge already acquired by previously fine-tuned models for distinct tasks. This gap has sparked considerable interest in developing methods for combining multiple models into a unified model Shu et al. (2021); Yang et al. (2022). Among these is model merging Yang et al. (2024), which presents a notable opportunity to fuse capabilities at low cost, such as the aTLAS method Zhang et al. (2024), which addresses the multi-source knowledge transfer for a new target task. It learns to scale and combine task vectors anisotropically Ilharco et al. (2022), which are the weight differences between fine-tuned models and their pre-trained state. The method operates by learning a distinct coefficient for each of the $T$ tasks, across each of the $L$ layers, and for each of $P$ partitions within a weight matrix. These coefficients collectively form a learned tensor with dimensions $T \times L \times P$, allowing for adjustments to the model's behavior for new tasks. While holding significant promise, aTLAS lacks mechanisms for granular parameter selection, which restricts the precision of knowledge fusion. Furthermore, aTLAS's memory footprint scales linearly with the number of added sources due to its reliance on using full task vectors. This design prevents the aggregation of larger models or a greater number

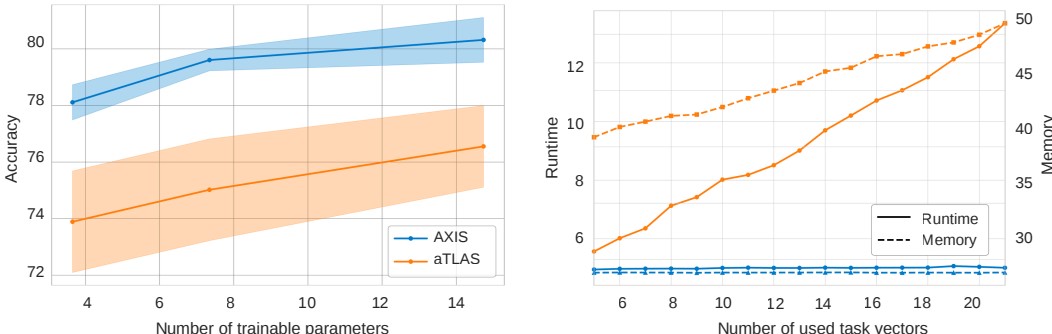

Figure 1: **Left**: Accuracy versus the number of trainable parameters for our method and aTLAS, averaged over all target tasks with ViT-B-32 architecture. Each data point corresponds to a fine-tuning parameter budget defined by the top N singular values (N=10%, 20%, and 40%). The solid line denotes the mean accuracy, while the shaded area represents the standard deviation. The variation is calculated over all source task vectors. **Right**: Scalability analysis for ViT-L-14 architecture with $N$=10% trainable parameters. As the number of source task vectors increases, the runtime and memory costs of aTLAS scale near-linearly. In contrast, our AXIS framework maintains a constant computational footprint.

of source models. As a result, its training is confined to multi-GPU environments, undermining its parameter-efficient benefits. This coarse-grained approach lacks a robust knowledge composition mechanism, making it susceptible to perturbations from both corrupted and pruned parameters and degraded inputs.

In this paper, we present a unified method that efficiently combines specialized knowledge from multiple fine-tuned source models in the parameter space to facilitate transfer to a new, unseen target task. We depart from the methodology proposed in the aTLAS paper, which assumes that the entire set of full-rank task vectors is used throughout the entire training process. Instead, we propose a more scalable approach that first aggregates knowledge and then allows for its efficient refinement during adaptation. First, we leverage Singular Value Decomposition (SVD) to decompose each task vector into its elementary, rank-one components. This allows us to identify and isolate granular patterns learned for each source task. A subsequent combination stage aggregates these components from all source models, performing a joint ranking to retain only a small, fixed number of the most significant ones. We term this strategy **AXIS**, as it embodies the principle of **A**ggregation by e**X**traction of **I**mportant **S**ingular components. Such selective aggregation ensures a stable memory usage and constant wall-time footprint during training, irrespective of the number of source models or original task matrix sizes (see Figure 1). Consequently, the proposed design is not only more parameter-efficient, but it also proves to be more robust. Our key contributions include:

- We introduce a scalable approach, AXIS, which outperforms the state-of-the-art method, aTLAS, across a wide spectrum of evaluation conditions, including 21 distinct tasks and various parameter budgets, covering three vision model scales as well as the language domain.

- The computational efficiency of AXIS is a key advantage, allowing for the scaling of knowledge transfer from a large number of source tasks and larger models.

- We demonstrate AXIS's robustness to parameter and input degradations, as well as its potential for transferring knowledge across models with diverse initializations.

- Through ablation studies, we offer insights into the underlying structure of knowledge composition and how it can be leveraged.

## 2 RELATED WORKS

**Model merging** is gaining traction as a promising approach to leverage fine-tuned models without requiring access to training data or incurring increased model size and inference costs. The merging stage itself demands low computational resources and could be entirely training-free. While substantial progress has been made in combining models with diverse architectures Du et al. (2025) or

those trained without a shared initialization Rinaldi et al. (2025); Stoica et al. (2023); Ainsworth et al. (2022), our work primarily focuses on distinct, yet highly prevalent paradigm where models originate from a common pre-trained base Akiba et al. (2025); Yang et al. (2023); Yadav et al. (2023). This shared origin allows for the direct application of task arithmetic Ilharco et al. (2022), enabling precise manipulation of weight differences to compose capabilities. Model merging can enhance single-task performance Wortsman et al. (2022a); Ramé et al. (2023); Jang et al. (2024) or be utilized in the creation of multitask models Marczak et al. (2025); Gargiulo et al. (2025). While merged models for multitask performance show limited promise for cross-domain compositional generalization Tam et al. (2024), we focus on explicitly reusing weights for distinct, new target tasks. Other prior works focus on merging reasoning skills with Chains-of-Thought Yin et al. (2025) for better zero-shot knowledge composition.

**Singular Value Decomposition (SVD)** offers a valuable approach for parameter-efficient fine-tuning (PEFT), allowing effective modifications within the eigenspectrum of pre-trained weights Wang et al. (2024); Bałazy et al. (2024); Peng et al. (2024); Meng et al. (2024). While many of these strategies achieve parameter efficiency by focusing on the singular values, diverse approaches exist Lingam et al. (2024). Others leverage SVD with reinforcement learning at inference time, adapting to unseen target tasks Sun et al. (2025). We introduce a unique adaptation strategy that diverges from prior work in two critical ways. First, we apply SVD to a multi-source merged model. Second, departing from the more varied heuristics seen before, our adaptation is guided exclusively by the largest singular values.

## 3 METHOD

### 3.1 PROBLEM STATEMENT

Let the parameters of the base, pre-trained model be denoted by $\theta_{\text{pre}}$. We consider a set of $T$ distinct tasks. For a given task $i$, the model is fine-tuned on a corresponding dataset $D_i$. The parameters of this fine-tuned model are denoted as $\theta_i$. Finally, the parameters for a specific layer $l$ within this model are represented by $\theta_i^{(l)}$. A task vector is the element-wise difference between the parameters of a fine-tuned model and its pre-trained counterpart. Building on this concept, we define a per-layer task difference to capture these modifications with greater granularity. Denoting the parameters of the base model for layer $l$ as $\theta_{\text{pre}}^{(l)}$ and the fine-tuned parameters for task $i$ at layer $l$ as $\theta_i^{(l)}$, we define **task vectors** $\tau_i^{(l)}$ as:

$$\tau_i^{(l)} = \theta_i^{(l)} - \theta_{\text{pre}}^{(l)}$$

For weight parameters that form a matrix (e.g., in linear or attention layers), we denote this difference specifically as the task matrix $\Delta_i^{(l)}$ to emphasize the structure suitable for SVD. For all other modules (e.g., biases, normalization), we retain the general term $\tau_i^{(l)}$. For these non-matrix parameters, we simply compute their element-wise average across all source tasks, similar to other works. The entire procedure, from decomposition to adaptation, is performed independently for each relevant layer in the model. For brevity, we will generally omit the layer index $(l)$. While non-parametric operations, such as activation functions, are applied during the model's forward pass, they do not have learnable weights and are therefore not represented in the task vector.

### 3.2 DECOMPOSING TASK MATRICES

To capture the structured modifications introduced by fine-tuning, we perform a granular analysis of each task matrix, $\Delta_i$, using Singular Value Decomposition (SVD). For a given task matrix $\Delta_i$ at any generic layer, we consider its SVD:

$$\Delta_i = \boldsymbol{U}_i \boldsymbol{\Sigma}_i \boldsymbol{V}_i^\top$$

where $U_i \in \mathbb{R}^{m \times r_i}$ and $V_i \in \mathbb{R}^{n \times r_i}$ are the matrices of left and right singular vectors, respectively, and $\Sigma_i \in \mathbb{R}^{r_i \times r_i}$ is a diagonal matrix containing the singular values $\sigma \in \mathbb{R}^{r_i}$. The value $r_i$ denotes the rank of the matrix $\Delta_i$ and corresponds to the number of its singular components.

Given a pre-trained model, parameterized by $\theta_{\text{pre}}$, and a library of $T - 1$ source task matrices, $\{\Delta_i\}_{i=1}^{T-1}$, our objective is to synthesize this knowledge to effectively adapt the model for a new,

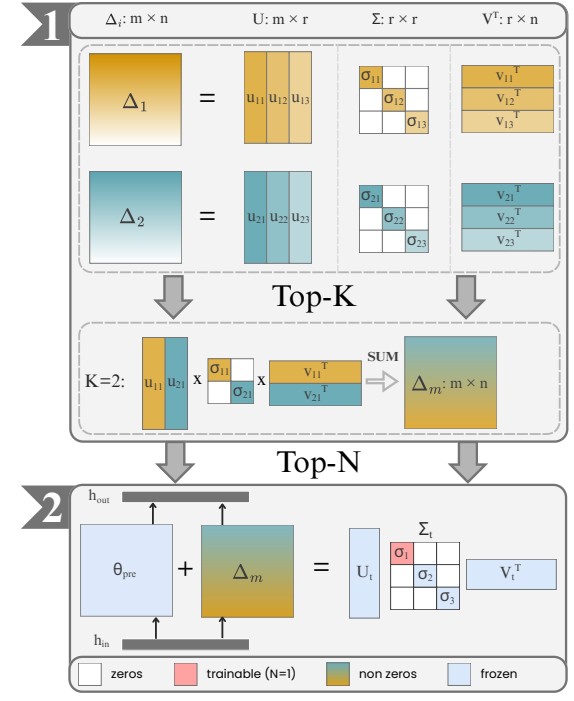

**Algorithm 1** AXIS
1: Initialize SVD components: $\mathcal{C} \leftarrow \emptyset$.
2: **for** each source task $i \in \{1, ..., T-1\}$ **do**
3:   Compute the SVD of $\Delta_i = U_i \Sigma_i V_i^\top$
4:   $\mathcal{C} \leftarrow \mathcal{C} \cup \{(\mathbf{u}_j, \sigma_j, \mathbf{v}_j^\top)\}_{j=1}^{r_i}$
5: **end for**
6: **Select** the top-K components to form $\mathcal{B}$
7:   $\text{Sort}_{\sigma_k\downarrow}(\mathcal{C}) \rightarrow \mathcal{B}$
8: **Assemble** non-orthogonal vectors:
9:   $U_m \leftarrow [u_1|u_2|\dots|u_K]$
10:   $\Sigma_m \leftarrow \text{diag}(\sigma_1, \sigma_2, \dots, \sigma_K)$
11:   $V_m \leftarrow [v_1|v_2|\dots|v_K]$
12: **Reconstruct** from components:
13:   $\Delta_m \leftarrow U_m \Sigma_m V_m^\top$
14: **Re-orthogonalize** the basis via SVD:
15:   $\Delta_m = U_t \Sigma_t V_t^\top$.
16: **Define** the set of learnable parameters $\Lambda$ as the top-$N$ singular values from $\Sigma_t$:
17:   $\Lambda \leftarrow [s_1, \dots, s_N]$.
18: **Define** frozen singular values:
19:   $\mathbf{s}_{\text{frozen}} \leftarrow \text{diag}(\Sigma_t) \setminus \Lambda$
20: **Reconstruct** with learned values:
21:   $\Delta_t \leftarrow U_t \text{diag}(\Lambda, \mathbf{s}_{\text{frozen}}) V_t^\top$.
22: **return** $\Delta_t$

Figure 2: An overview of the AXIS framework. The process consists of two stages: **(1) Extraction and aggregation:** Each source task matrix $(\Delta_1, \Delta_2, \dots)$ is decomposed into its elementary singular components using SVD. The most salient components from all sources are selected based on a global Top-K ranking of their singular values. These K components are then summed to synthesize the merged task matrix, $\Delta_m$. For clarity, the diagram illustrates this with K = 2. **(2) Adaptation:** To form a stable and decorrelated transfer basis, $\Delta_m$ is re-parameterized via a final SVD. The model is then adapted to the target task by fine-tuning only a small subset (Top-N) of the most principal singular values of the resulting matrix $\Sigma_t$ in each layer.

unseen target task. The original training datasets for these source tasks, i.e., $\{D_1, ..., D_{T-1}\}$, are not available. For the target task, we only have access to its labeled dataset, which is partitioned into a training set $D_t^{\text{train}}$ and a test set $D_t^{\text{test}}$.

### 3.3 OUR TWO-STAGE COMPOSITION FRAMEWORK

STAGE 1: KNOWLEDGE EXTRACTION AND AGGREGATION.

Our core hypothesis is that the most transferable useful knowledge for the target task, encoded across diverse source tasks $\{\Delta_i\}_{i=1}^{T-1}$, is within the principal singular components, which represent the most dominant structural patterns in the parameter space. Therefore, for each source task matrix $\Delta_i$, we perform SVD to decompose it into a set of orthogonal components. Each component is a triplet $(\mathbf{u}_{i,j}, \sigma_{i,j}, \mathbf{v}_{i,j}^\top)$, where $j$ is the component index for a given task $i$. Consequently, we propose an aggregation strategy based on a global ranking of all components from all source task matrices. We then select the Top-K components with the highest singular values to construct the transfer basis:

$$\mathcal{B} = \{(\mathbf{u}_k, \sigma_k, \mathbf{v}_k^\top)\}_{k=1}^K, \text{ where } \sigma_k \geq \sigma_{k+1}, \forall k$$

Finally, the merged task matrix, $\Delta_m$, is synthesized by summing the Top-$K$ selected rank-one components:

$$\Delta_m = \sum_{k=1}^K \mathbf{u}_k \sigma_k \mathbf{v}_k^\top.$$

By prioritizing these high-magnitude components, we aim to build a new, effective pre-trained state for any unknown downstream task. We empirically validate the quality of the merged model and the component selection strategy against alternatives in our ablation studies.

STAGE 2: TARGET TASK ADAPTATION.

In the second stage, the merged knowledge $\Delta_m$ is adapted to the specific target task. We define the final target task vector $\Delta_t$ as a function of $\Delta_m$ and a small set of *learnable parameters* $\Lambda$ that minimize the cross-entropy loss $\mathcal{L}$ on the target dataset:

$$\Lambda^* = \underset{\Lambda}{\operatorname{argmin}} \, \mathbb{E}_{(x,y) \in D_t} \left[ \mathcal{L} \left( f(x; \theta_{\text{pre}} + \Delta_t(\Lambda)), y \right) \right]$$

For a parameter-efficient adaptation, we apply gradient-based learning exclusively to the top-$N$ singular values of $\Delta_t$, which constitute the set $\Lambda$. The remaining singular vectors and less significant components are kept frozen. The resulting full model parameters for the target task are $\theta_t = \theta_{\text{pre}} + \Delta_t(\Lambda)$ and the full, step-by-step process is formalized in Algorithm 1 and Figure 2.

The synthesized matrix $\Delta_m$ represents a rich but intermediate consolidation of knowledge from multiple source tasks. To transform this aggregation into a computationally stable and effective basis for adaptation, we re-parameterize it using a final SVD. This procedure, $\Delta_m \rightarrow U_t \Sigma_t V_t^\top$, serves a dual purpose. First, it constructs a new set of orthogonal vectors, $U_t$ and $V_t$, creating a decorrelated basis that optimally represents the merged transformation in the sense of the Frobenius norm. Second, it yields a new diagonal matrix $\Sigma_t$, whose values reflect the true importance of the components within the combined matrix $\Delta_m$ and also serve as the isolated set of learnable parameters, $\Lambda$, for the subsequent fine-tuning.

# 4 RESULTS

## 4.1 EXPERIMENTAL SETUP

To evaluate the performance, scalability, and robustness of our method, we benchmark it against the recent state-of-the-art method, aTLAS, which serves as our baseline. The experimental framework is based on diverse image classification tasks, including texture recognition (DTD), satellite imagery (EuroSAT), and fine-grained visual categorization (Flowers102). The experimental setup employs a leave-one-out protocol. For each target task, we incrementally aggregate knowledge assets by varying the number of source task vectors from one up to the maximum of $T-1$ in a fixed, predefined sequence. By default, we use the pre-trained Vision Transformer (ViT-B-32) variant of the CLIP model Radford et al. (2021). Our primary performance metric is the Top-1 accuracy evaluated on the test set of each target task. All results are presented under a matched number of trainable parameters and within the range used by aTLAS method. Our evaluation adapts the comprehensive benchmark, publicly released task vectors, and training protocols established by the authors of aTLAS to ensure a direct and fair comparison. For each target task adaptation, the fine-tuning process utilizes the complete, standard training set. To provide a one-to-one comparison, we adopted the same training configuration used for the aTLAS baseline and ran all its experiments within this consistent framework. Specifically, each adaptation runs for 10 epochs with a learning rate of $10^{-1}$. All setup details and results with seven textual datasets from Yadav et al. (2023) utilizing the T5-Base language model (see Figure 10) are provided in the Appendix.

## 4.2 PERFORMANCE AND EFFICIENCY GAINS OVER ATLAS

For each target task, we incrementally build the merged task vector, $\Delta_{target}$, by aggregating an increasing number of source task vectors. For example, a single model synthesized from 16 source vectors is then independently fine-tuned 21 times - once for each distinct target task as illustrated in Figure 3. This entire process is repeated for every aggregation level, and the outcomes are averaged to produce the final performance curves. The parameter budgets $N$ of 10%, 20%, and 40% are determined by the percentage of trainable singular values selected from each task matrix; their sum across all matrices results in total trainable parameter counts of approximately 3.6k, 7.3k, and 14.7k, respectively, in the ViT-B-32 version. The results demonstrate that our approach outperforms aTLAS across the entire spectrum of source task quantities on both the ViT-B-32 (illustrated in Figure 3) and ViT-L-14 architectures (see Figure 9 in the Appendix).

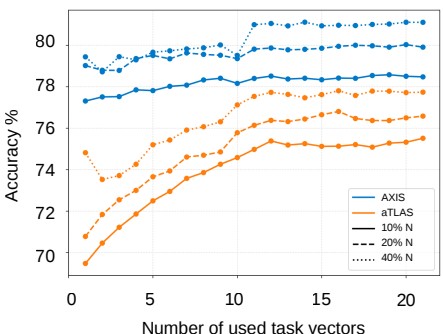 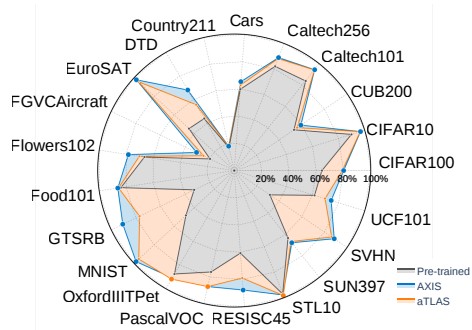

Figure 3: **Left:** Performance comparison with aTLAS varying the number of trainable parameters with the ViT-B-32 architecture. Each point represents a model configuration that was independently adapted to all target tasks. The plotted value is the mean performance across these tasks. **Right:** Detailed per-task comparison of the merged models (AXIS vs. aTLAS) utilizing 16 task vectors. To clarify performance differences in overlapping regions, the marker of the superior method for a given task is rendered on top. Overall, AXIS achieves a higher average accuracy of 78.42% compared to 75.13% for aTLAS.

Our method shows higher parameter efficiency, as illustrated in Figure 1. The figure compares AXIS with aTLAS, showing that for any given parameter budget, our approach yields higher average accuracy. Furthermore, the noticeably smaller shaded area for AXIS indicates a lower standard deviation, highlighting that our aggregation mechanism is more stable and less sensitive to variations in the number of source task vectors used.

**Memory and Runtime Scalability.** A key advantage of our method is its significantly lower computational overhead compared to baselines like aTLAS. The memory and runtime costs of aTLAS scale near-linearly with the number of source models, as it learns a distinct coefficient for each of the $T$ source tasks across every layer and parameter partition $P$ during the fine-tuning process. This means that all source task vectors must be present in memory throughout the entire adaptation phase for a new target task.

In stark contrast, AXIS decouples the process into two distinct stages. The first stage, knowledge aggregation, is a fast, one-time operation. It efficiently processes all $T - 1$ source task vectors using SVD and consolidates them into a single, fixed-size merged matrix, $\Delta_m$. The subsequent, and most computationally intensive, fine-tuning stage operates only on this compact $\Delta_m$. As a result, the memory footprint and runtime of the adaptation phase remain constant, regardless of the number of source models initially aggregated. This design choice not only makes our approach more scalable but also significantly reduces the resources required for fine-tuning, as is illustrated in Figure 1.

### 4.3 ROBUSTNESS TO NOISE AND SPARSITY IN SOURCE PARAMETERS

To evaluate the robustness of our method with unreliable Li et al. (2025) or compressed Iurada et al. (2025); Li et al. (2025) source task vectors, we designed two specific scenarios. The first simulates contamination from a single, low-quality source, for instance, due to training instabilities. The second scenario evaluates how effectively these approaches leverage knowledge when all source task vectors are heavily pruned. Both investigations explore the method's capacity to merge a more diverse and challenging spectrum of models, expanding its practical applicability.

We formed aggregations of source task vectors of varying sizes, ranging from three to eight, to demonstrate the effect of a single faulty source. In each aggregation, one task vector was intentionally corrupted, while the others remained intact. The corruption was applied by adding zero-mean Gaussian noise to the weights of an original task vector. To ensure a significant level of disruption, the standard deviation of the noise was scaled to 50% of the Frobenius norm of that task matrix ($\sigma = 0.5 \cdot ||\Delta_i||_F$). The results illustrated in Figure 4 demonstrate that while both methods experience some performance degradation in the presence of a corrupted source, the impact on our method is significantly less pronounced. This indicates a more robust knowledge transfer mechanism. We

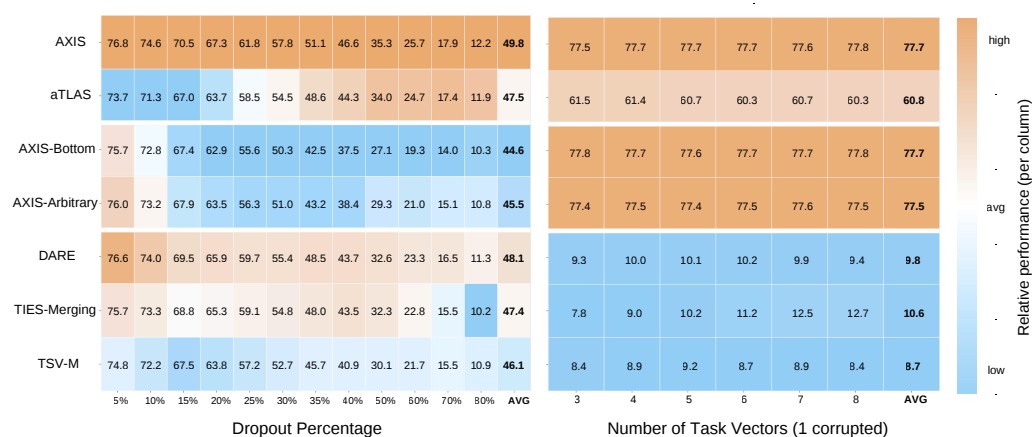

Figure 4: **Left:** The heatmap illustrates the average accuracy across all target tasks. Results indicate that AXIS outperforms the baselines under challenging conditions where input information is partially hidden, with up to 80% of patches masked. **Right:** Results averaged across all target tasks summarize the robustness to a single corrupted source task vector (out of 3 to 8 total). Our method, AXIS, demonstrates superior resilience to this scenario compared to aTLAS and other merging methods.

observe that our SVD-based selection process, by focusing on components with the highest singular values, is less susceptible to the unstructured perturbations introduced into a single source vector.

To assess the robustness of our method from a compression perspective, each of the source task vectors underwent magnitude-based pruning (see Figure 20). We applied a high-level ratio, ensuring that specialized knowledge was not catastrophically degraded. The subsequent analysis suggests that our approach can more effectively leverage the knowledge contained within highly sparse task vectors, showcasing a distinct advantage in utilizing compressed knowledge.

### 4.4 ROBUSTNESS TO INPUT DATA DEGRADATION

Building on findings that merging models fine-tuned with distinct hyperparameters on the same task leads to greater stability under distribution shifts Wortsman et al. (2022a;b), we explore whether aggregating knowledge from multiple, diverse models, each fine-tuned with the same set of hyperparameters, can similarly construct a more robust representation. For this experiment, the AXIS and aTLAS models were built by aggregating the complete set of $T - 1$ source task vectors and fine-tuning them for each target task.

The model's accuracy on images with randomly omitted patches can serve as a direct test, which was previously used to measure model robustness Paul & Chen (2022) or ability to perform prediction with partial information Pardyl et al. (2025), providing unique insight into a model's internal representation, as this form of robustness is often less correlated with baseline model performance than other image perturbations Malik et al. (2025). To ensure a fair comparison, a fixed seed guarantees that all methods are evaluated using the same masked patches for each dropout level. In Figure 4, AXIS shows resilience when almost all complete information is available, and degrades more slowly as input degradation becomes more severe. This capability is essential for real-world scenarios with incomplete data and follows prior research aimed at improving model resilience to partial visual information Liu et al. (2023); Tang et al. (2022) (see Table 8). Additionally, we demonstrate better robustness capabilities of AXIS than aTLAS against a set of 12 common image corruptions Hendrycks & Dietterich (2019) with five severity levels in the Appendix (see Figure 16 and Figure 17).

## 5 ANALYSIS

### 5.1 BROAD COMPARISON

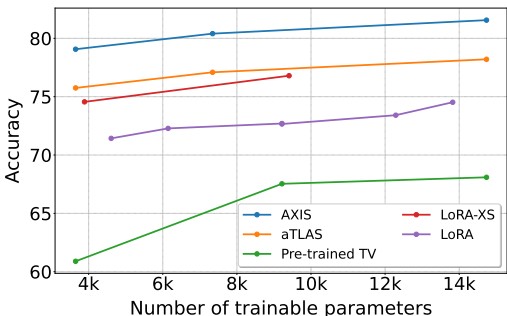

| Method | N = 10% | N = 20% | N = 40% |
|---|---|---|---|
| DARE + Stage 2 | 78.09 ± 0.06 | 79.69 ± 0.04 | 80.77 ± 0.09 |
| Average + Stage 2 | 78.19 ± 0.15 | 79.45 ± 0.15 | 79.43 ± 0.70 |
| TIES + Stage 2 | 77.39 ± 0.03 | 78.99 ± 0.05 | 80.27 ± 0.05 |
| TSV-M + Stage 2 | 76.41 ± 0.05 | 78.69 ± 0.07 | 80.41 ± 0.11 |
| aTLAS | 75.50 ± 0.03 | 75.93 ± 0.44 | 77.66 ± 0.05 |
| **AXIS** | **78.46 ± 0.04** | **79.93 ± 0.11** | **81.13 ± 0.07** |

Table 1: Performance comparison with aTLAS and merging methods when followed by our Stage 2 adaptation. While the best results are obtained by AXIS, the adaptation mechanism itself is a potent and versatile tool for refining diverse multi-capability models. Crucially, while other merging baselines achieve lower, but competitive accuracy, AXIS exhibits significantly superior robustness against weights corruption (see Figure 4). All results are averaged over 3 seeds.

Figure 5: Performance comparison with competing methods, including PEFT variants. The proposed merge-and-tune paradigm in AXIS achieves a more efficient performance-parameter trade-off.

To demonstrate the advantages of our approach, we compare it with different finetuning methods, in particular with PEFT methods. This includes the widely-adopted LoRA Hu et al. (2022) and its enhanced variant LoRA-XS Bałazy et al. (2024). Additionally we try to further adapt the pre-trained weights as a single task vector. As the Fig 5 shows, our method efficiently outperform these techniques, effectively reusing already finetuned weights.

We further take inspiration from model merging techniques and ask the question whether a general, multi-task model serve as an effective knowledge base for our Stage 2 adaptation? To test this hypothesis, we substitute our AXIS aggregation with several established multi-task merging techniques, such as DARE Yu et al. (2024), TIES-Merging Yadav et al. (2023), TSV-M Gargiulo et al. (2025) and simple averaging, treating their merged weights as alternative initializations. As the results in Table 1 demonstrate, these multi-task models indeed form a potent foundation for our adaptation mechanism, albeit slightly below the performance of the AXIS method. This suggests that the Stage 2 is not rigidly dependent on a single aggregation method but can effectively refine knowledge from various merged, multi-capability models. Crucially, these alternative merging baselines lack the structural robustness inherent in our approach, as illustrated in Figure 4. They exhibit significant performance degradation in realistic scenarios involving partial input information or corrupted source models, whereas AXIS maintains superior resilience.

## 5.2 SCALABILITY AND PARAMETER SENSITIVITY OF AXIS

To assess the sensitivity of our method to the size of the transfer basis, we conducted an ablation study on the number of selected components, $K$. This sole hyperparameter directly controls the dimensionality of the aggregated knowledge consolidated into the merged task matrix, $\Delta_m$. In this experiment, we varied the value of $K$ used in our *top components* aggregation strategy, where components from all source tasks are globally ranked by their singular values before the top $K$ are selected to form the transfer basis. Our default choice of $K = 76$ (approximating 10% of each layer's rank) proves to be a robust heuristic. The plot demonstrates that performance remains high, with the drop being less than 1.5% even for large $K$ (Figure 6). Overall, we find that limiting $K$ to be less than 20% of total parameters provides robust results. We hypothesize that including additional components may introduce more task-specific details, which are not necessarily important for the target task (see Figure 11 and Figure 12).

Additionally, we evaluate how scaling the number of trainable parameters ($N$) affects model performance (Figure 7). Specifically, setting $N = 100\%$ corresponds to fine-tuning all the singular values in each AXIS task matrix. The improvements begin to diminish once $N$ exceeds 60%. Thus, N serves as a control parameter that balances computational requirements and final performance.

## 5.3 COMPONENTS SELECTIONS STRATEGY

To evaluate the quality of component aggregation, we test three selection criteria from a global pool of all aggregated SVD components. We compare the impact of selecting components with

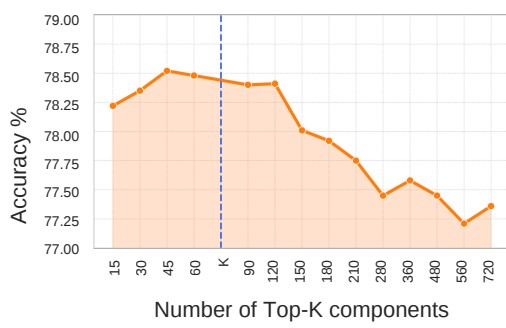

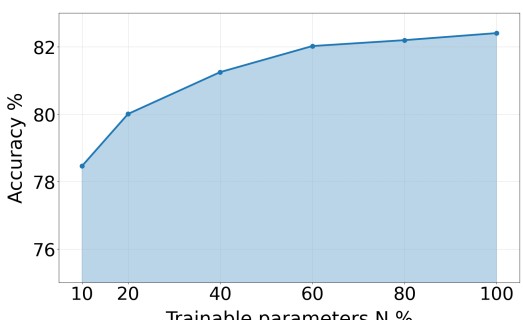

Figure 6: Performance sensitivity to the number of aggregated components $K$. Best averaged results are obtained for $K < 20\%$. The default setting is 10% ($K = 76$).

Figure 7: The AXIS scales consistently with the number of trained parameters ($N\%$), showing improved performance as $N$ increases, with gains tapering off beyond 60%.

| N (%) | Top | Arbitrary | Bottom |
|---|---|---|---|
| 10 | $78.46 \pm 0.04$ | $77.83 \pm 0.04$ | $77.56 \pm 0.02$ |
| 20 | $79.93 \pm 0.11$ | $79.79 \pm 0.03$ | $79.81 \pm 0.05$ |
| 40 | $81.13 \pm 0.07$ | $81.17 \pm 0.08$ | $81.13 \pm 0.04$ |

Table 2: Performance comparison of different SVD component selection strategies within the AXIS framework, demonstrating their comparable effectiveness. However, the choice of components is crucial for ensuring the resilience of the model, as illustrated in the Figure. 4

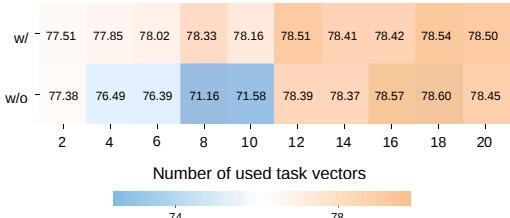

Figure 8: Skipping the final SVD orthogonalization results in a decline in performance, especially when combining a moderate number of task vectors.

the highest singular values (*top components*), the lowest (*bottom components*), and those chosen arbitrarily (*arbitrary components*). The results of this comparison are presented in Table 2, which indicates that the *top components* strategy yields the best performance. While selecting the *top components* components yields the highest accuracy, this advantage is most pronounced at lower parameter budgets. As the number of trainable parameters increases, the performance of all three strategies converges, suggesting that the importance of the initial component selection decreases as the model is given more trainable parameters. However, this convergence does not extend to robustness; see Figure 4. In the Appendix, we provide additional evidence demonstrating that the top components strategy captures the most transferable structural knowledge.

## 5.4 STABILIZING THE TRANSFER BASIS

Instead of performing the final SVD re-parameterization, the layer's weights were reconstructed directly from the aggregated components $\Delta_m$. For our primary strategy of top component selection, this omission results in significant performance degradation when a moderate number of task vectors are aggregated (Figure 8).

## 6 CONCLUSION

We presented AXIS, a framework that addresses multi-source knowledge transfer through the extraction, aggregation, and adaptation of useful knowledge for the target task. Furthermore, the framework enables efficient final adaptation while demonstrating robustness to degradations at both the parameter and input levels. Although the AXIS assumes a shared architecture and pre-training origin, our experiments in the Appendix demonstrate remarkable robustness to deviations from these constraints. This enables effective knowledge transfer even across models with different initializations and architectural variants with different scales.

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

# A  APPENDIX

OVERVIEW

The appendix provides supplementary material to support and expand upon the main findings of our paper. Additionally, we provide code in the supplementary material. To ensure clarity the contents are organized as follows:

- **Evaluation Protocol in Vision Benchmark:** We begin by providing comprehensive details on the evaluation protocol and datasets used throughout our experiments.

- **Performance on ViT-L-14 Architecture:** We present a comparative performance analysis of AXIS and aTLAS using the larger ViT-L-14 architecture, demonstrating that the advantages of our method scale effectively to more powerful models.

- **Performance on T5-Base Language Models:** We demonstrate the generalization of AXIS beyond computer vision by evaluating its performance on the T5-Base architecture across seven natural language processing tasks.

- **Validation of Adaptation Flexibility:** We empirically validate the efficiency of optimizing singular values by comparing it against alternative SVD-based strategies.

- **Investigating Transfer Boundaries:** We explore the limits of our framework by applying AXIS to cross-architecture and cross-initialization scenarios.

- **Transferability by SVD Components:** We provide an in-depth analysis justifying our ranking strategy, showing that high-magnitude singular components consistently align best with the ground-truth target task direction.

- **AXIS Performance at Full Parameter Budget:** We analyze the model's behavior as the number of trainable parameters budget approaches 100%.

- **Multi-Task Performance of Axis:** We evaluate a joint training strategy for simultaneous adaptation to multiple tasks.

- **Zero-Shot Transferability of Trained Models:** We assess the cross-task generalization of adapted models.

- **Dynamic Top-K Selection:** We benchmark our fixed $K$ component selection strategy against an dynamic method.

- **Incremental Knowledge Aggregation:** We introduce a streaming aggregation protocol that updates the model sequentially.

- **In-depth Robustness Analyses:** We conduct a series of thorough evaluations to validate the robustness of our framework under challenging conditions. These include:
  - Resilience to 12 common image corruptions across five distinct severity levels.
  - Performance evaluation across different levels of training data availability for the target task.
  - Robustness against altered source parameters, including scenarios with noisy or heavily pruned task vectors.

- **Component Selection:** We present a detailed ablation study comparing our default component aggregation strategy (top components) against a range of alternative methods.

- **Impact of Final SVD:** We provide details of the role of the final SVD re-parameterization step in stabilizing the transfer basis across a couple of selection strategies.

- **Detailed Main Results:** We then provide extensive results with the ViT-B-32 architecture. These tables offer a granular performance breakdown, detailing per-target-task accuracy for different numbers of aggregated source task vectors and varying budgets of trainable parameters ($N$).

## A.1  EVALUATION PROTOCOL

To ensure a direct and fair comparison, we adopt the comprehensive benchmark, publicly released task vectors, and training protocols established by the authors of aTLAS. Their framework

| Dataset | Classes | Splits | | | Epochs | Fine-tuned accuracy (%) | |
|---------|---------|--------|-----|------|--------|-------------|----------|
| | | train | val | test | | ViT-B/32 | ViT-L/14 |
| Cars | 196 | 7,330 | 814 | 8,041 | 35 | 78.26 | 91.67 |
| DTD | 47 | 3,384 | 376 | 1,880 | 76 | 78.94 | 84.73 |
| EuroSAT | 10 | 21,600 | 2,700 | 2,700 | 12 | 98.89 | 99.81 |
| GTSRB | 43 | 23,976 | 2,664 | 12,630 | 11 | 99.14 | 99.30 |
| MNIST | 10 | 55,000 | 5,000 | 10,000 | 5 | 99.65 | 99.77 |
| RESISC45 | 45 | 17,010 | 1,890 | 6,300 | 15 | 95.94 | 97.14 |
| SUN397 | 397 | 17,865 | 1,985 | 19,850 | 14 | 75.40 | 81.98 |
| SVHN | 10 | 68,257 | 5,000 | 26,032 | 4 | 97.38 | 97.97 |
| CIFAR10 | 10 | 45,000 | 5,000 | 10,000 | 5 | 98.05 | 99.22 |
| CIFAR100 | 100 | 45,000 | 5,000 | 10,000 | 6 | 89.09 | 93.01 |
| ImageNet | 1,000 | 1,276,167 | 5,000 | 50,000 | 10 | 76.41 | 85.52 |
| STL10 | 10 | 4,500 | 500 | 8,000 | 4 | 98.55 | 99.62 |
| Food101 | 101 | 70,750 | 5,000 | 25,250 | 15 | 88.68 | 95.37 |
| Caltech101 | 101 | 6,941 | 694 | 1,736 | 10 | 94.41 | 94.82 |
| Caltech256 | 257 | 22,037 | 2,448 | 6,122 | 8 | 92.60 | 97.17 |
| FGVCAircraft | 100 | 3,334 | 3,333 | 3,333 | 60 | 40.65 | 68.11 |
| Flowers102 | 102 | 1,020 | 1,020 | 6,149 | 40 | 90.08 | 97.84 |
| OxfordIIITPet | 37 | 3,312 | 368 | 3,669 | 5 | 92.15 | 95.91 |
| CUB200 | 200 | 5,395 | 599 | 5,794 | 20 | 73.56 | 86.35 |
| PascalVOC | 20 | 7,844 | 7,818 | 14,976 | 10 | 88.42 | 92.05 |
| Country211 | 211 | 31,650 | 10,550 | 21,100 | 15 | 21.99 | 38.06 |
| UCF101 | 101 | 7,639 | 1,898 | 3,783 | 20 | 85.01 | 92.55 |

Table 3: Comparison of full fine-tuning model accuracy per dataset

provides task vectors obtained by fine-tuning the pre-trained CLIP Radford et al. (2021) model on distinct image recognition datasets: Stanford Cars Krause et al. (2013), DTD Cimpoi et al. (2014), EuroSAT Helber et al. (2019), GTSRB Stallkamp et al. (2011), MNIST LeCun (1998), RESISC45 Cheng et al. (2017), SUN397 Xiao et al. (2016), SVHN Netzer et al. (2011), CIFAR10 Krizhevsky et al. (2009), CIFAR100 Krizhevsky et al. (2009), ImageNet Russakovsky et al. (2015), STL10 Coates et al. (2011), Food101 Bossard et al. (2014), Caltech101 Fei-Fei et al. (2006), Caltech256 Griffin et al. (2007), FGVCAircraft Maji et al. (2013), Flowers102 Nilsback & Zisserman (2008), Oxford Pets Parkhi et al. (2012), CUB200 Welinder et al. (2010), PascalVOC Everingham et al. (2015), Country211 Radford et al. (2021), and UCF101 Soomro et al. (2012). The original fine-tuning for these vectors was performed using the AdamW optimizer Loshchilov & Hutter (2017) with a learning rate of $10^{-5}$, a batch size of 128, and a weight decay of 0.1 for the ViT-B-32 architecture. Table 3 provides dataset details, their corresponding hyperparameters, and the fine-tuning accuracy achieved with full-finetuning.

During the target task adaptation stage, we fine-tune the merged model for each dataset independently, using the same hyperparameters as the aTLAS baseline (each adaptation runs for 10 epochs with a learning rate of $10^{-1}$). The batch size is adjusted based on the model architecture: 64 for the ViT-B-32 and ViT-B-16 model and 128 for the larger ViT-L-14 model. For the ViT-L-14 architecture, both methods originally use two steps of gradient accumulation. To ensure a controlled and reproducible evaluation provided by aTLAS, the source task vectors are aggregated incrementally in a fixed, pre-defined sequence. The order of aggregation is as follows: Cars, DTD, EuroSAT, GTSRB, MNIST, RESISC45, SUN397, SVHN, CIFAR10, CIFAR100, ImageNet, STL10, Food101, Caltech101, Caltech256, FGVCAircraft, Flowers102, OxfordIIITPet, CUB200, PascalVOC, Country211, and UCF101. Each experimental run was conducted once with a single random seed across our comprehensive evaluation, which included 21 target tasks, multiple aggregation levels, and varying parameter budgets.

### A.1.1 COMPUTATIONAL ENVIRONMENT

All experiments were conducted within a high-performance computing (HPC) cluster equipped with a heterogeneous GPU environment. The available resources included partitions with NVIDIA RTX 4090, NVIDIA V100, and NVIDIA A100 GPUs. The results reported in this paper, generated using the ViT-L-14 architecture, were obtained with nodes equipped with NVIDIA A100-SXM4-80GB GPUs. Our software stack was built upon the CUDA 12.2 toolkit with NVIDIA driver version 535.183.01.

### A.2 PERFORMANCE ON VIT-L-14 ARCHITECTURE

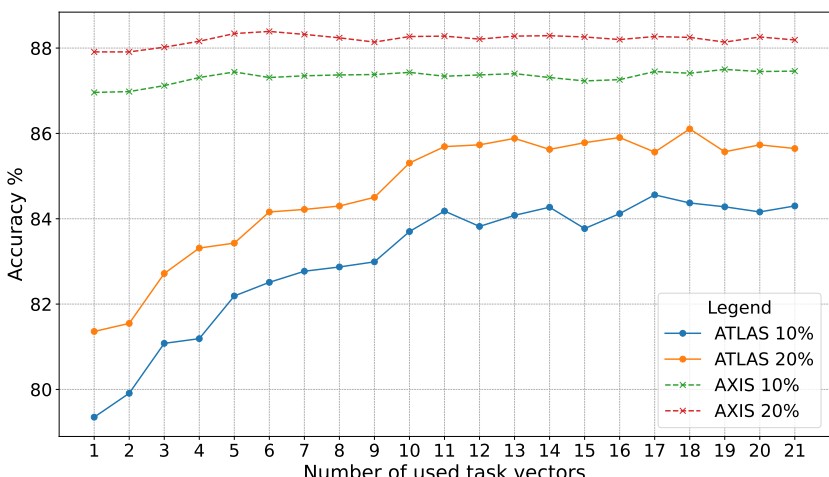

Figure 9: AXIS outperforms aTLAS on the ViT-L-14 architecture with $N = 10\%$ and $N = 20\%$ of trainable singular values. Each point is the mean accuracy across 21 independently evaluated target tasks. The plot illustrates the accuracy gain as the number of aggregated source tasks increases.

To validate the scalability and effectiveness of our approach on larger models, we replicated our experiments using the ViT-L-14 architecture. The results demonstrate the advantages of the AXIS framework. The performance comparison for the $N = 10\%$ and $N = 20\%$ parameter budget is illustrated in Figure 9, where AXIS consistently outperforms aTLAS as the number of aggregated source tasks increases. Further analysis across different parameter budgets confirms these findings.

| Dataset | AXIS (%) | aTLAS (%) | Absolute gain (AXIS) |
|---|---|---|---|
| CIFAR100 | 80.13 | 79.14 | +0.99 |
| CIFAR10 | 96.78 | 96.51 | +0.27 |
| CUB200 | 59.06 | 55.82 | +3.24 |
| Caltech101 | 94.47 | 94.18 | +0.29 |
| Caltech256 | 88.88 | 88.09 | +0.79 |
| Cars | 65.30 | 62.82 | +2.48 |
| Country211 | 18.28 | 18.08 | +0.20 |
| DTD | 68.09 | 55.69 | +12.40 |
| EuroSAT | 97.74 | 95.67 | +2.07 |
| FGVCAircraft | 30.63 | 24.96 | +5.67 |
| Flowers102 | 78.37 | 70.24 | +8.13 |
| Food101 | 86.23 | 85.63 | +0.60 |
| GTSRB | 90.58 | 76.96 | +13.62 |
| MNIST | 98.03 | 95.30 | +2.73 |
| RESISC45 | 87.76 | 78.95 | +8.81 |
| SUN397 | 67.62 | 66.50 | +1.12 |
| SVHN | 88.65 | 86.32 | +2.33 |
| UCF101 | 74.31 | 69.60 | +4.71 |
| OxfordIIITPet | 90.79 | 91.82 | -1.03 |
| PascalVOC | 87.16 | 87.18 | -0.02 |
| STL10 | 97.91 | 98.33 | -0.42 |
| **Average** | **78.42** | **75.13** | **+3.29** |

Table 4: The table clarifies that aTLAS holds a marginal advantage on only 3 datasets (OxfordII-ITPet, PascalVOC, STL10), with two of these (PascalVOC, STL10) being statistically negligible, likely falling within the variance of a single-seed run. This superior per-task performance with ViT-B-32 architecture is visually detailed in the spider plot on the right side of Figure 3.

## A.3 PERFORMANCE ON T5-BASE LANGUAGE MODELS.

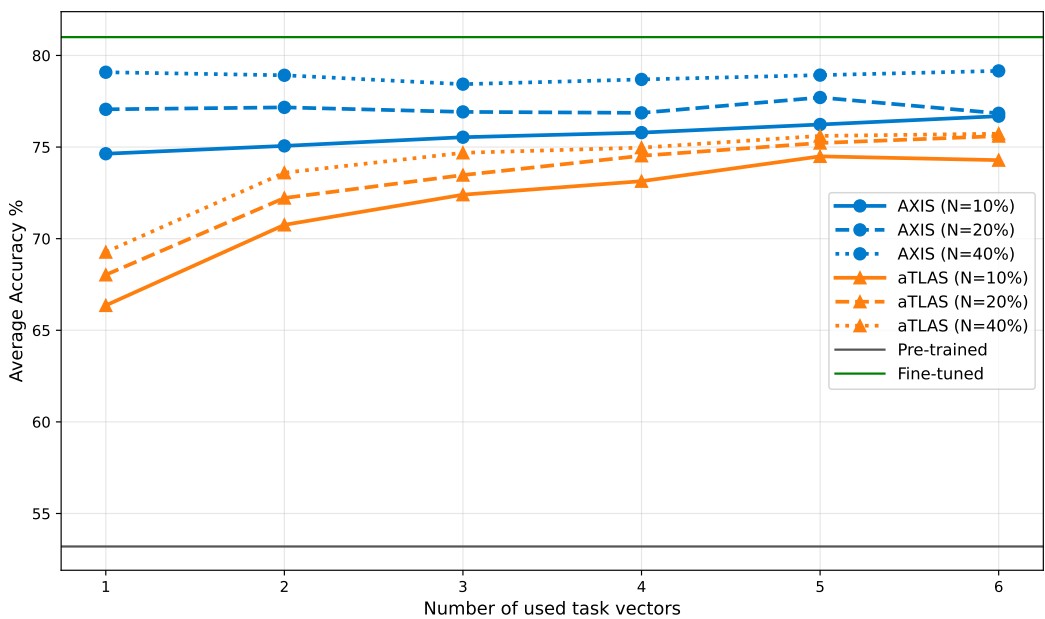

Figure 10: AXIS consistently outperforms the aTLAS baseline across seven diverse NLP benchmarks under varying source task aggregation levels and parameter budgets.

We extend the evaluation of the AXIS framework to the language domain using the T5-base architecture, adopting the multi-task merging protocol established in TIES-Merging. The evaluation encompasses seven NLP datasets: question answering (QASC Khot et al. (2020), WikiQA Yang et al. (2015), and QuaRTz Tafjord et al. (2019)), Paraphrase Identification (PAWS Zhang et al. (2019)), Sentence Completion (Story Cloze Sharma et al. (2018)), and Coreference Resolution (Winogrande Sakaguchi et al. (2021) and WSC Levesque et al. (2012)).

Figure 10 reports the average performance for an increasing number of aggregated source models, denoted by $s$ (ranging from 1 to $T - 1$). To ensure statistical robustness and mitigate selection bias, we performed an exhaustive evaluation of all valid source subsets. Specifically, for any given target task with a pool of $T - 1 = 6$ available sources, we averaged the results across all possible combinations for each subset size $s$. This entailed computing the mean performance over $\binom{6}{1} = 6$, $\binom{6}{2} = 15$, $\binom{6}{3} = 20$, $\binom{6}{4} = 15$, $\binom{6}{5} = 6$, and $\binom{6}{6} = 1$ distinct source combinations per target task. The final results are averaged over all target tasks. We benchmark AXIS across three distinct trainable parameter budgets, defined by the percentage of fine-tuned singular values ($N \in \{10\%, 20\%, 40\%\}$). For a direct and fair comparison of efficiency, the aTLAS baseline was evaluated using a matching budget of trainable parameters. The empirical results demonstrate that AXIS consistently outperforms the baseline across all aggregation levels, confirming that the method's efficacy in multi-source knowledge transfer generalizes beyond the vision domain.

| Dataset | AXIS (Tuning $\Sigma_t$ Diag) | A. FT in Singular Vectors | B. FT in $\Sigma_t$ Random |
|---|---|---|---|
| CIFAR10 | 98.29% | 92.05% | 97.92% |
| CIFAR100 | 86.61% | 66.96% | 84.31% |
| CUB200 | 70.85% | 34.93% | 65.17% |
| Caltech101 | 96.89% | 81.11% | 96.14% |
| Caltech256 | 93.24% | 78.05% | 91.56% |
| Cars | 79.22% | 48.35% | 75.04% |
| Country211 | 24.84% | 13.36% | 21.01% |
| DTD | 73.35% | 31.28% | 68.40% |
| EuroSAT | 98.48% | 72.00% | 98.78% |
| FGVCAircraft | 40.11% | 14.10% | 38.07% |
| Flowers102 | 86.03% | 29.65% | 72.08% |
| Food101 | 91.94% | 81.29% | 90.19% |
| GTSRB | 95.30% | 43.95% | 94.75% |
| MNIST | 99.04% | 85.88% | 99.05% |
| OxfordIIITPet | 94.85% | 77.00% | 93.59% |
| PascalVOC | 89.82% | 45.83% | 88.43% |
| RESISC45 | 94.08% | 54.43% | 93.10% |
| STL10 | 99.31% | 89.78% | 98.78% |
| SUN397 | 74.01% | 46.19% | 70.92% |
| SVHN | 95.19% | 77.77% | 94.81% |
| UCF101 | 84.93% | 41.69% | 80.49% |
| **Average** | **84.11%** | **57.41%** | **81.55%** |

Table 5: Comparison of adaptation strategies of Stage 2 across all target tasks for the ViT-B-16 model under different fine-tuning (FT) strategies. Our method (AXIS) outperforms alternative tuning strategies while maintaining an identical parameter budget.

## A.4 VALIDATION OF ADAPTATION FLEXIBILITY

To validate the hypothesis that tuning principal singular values does not limit adaptation flexibility, we conducted two comparative experiments while maintaining an identical budget of trainable parameters ($N$):

1. **Values vs. Vectors:** We compared our method (tuning values, freezing vectors) against an alternative approach where the singular values are frozen, and a random subset of parameters within the singular vectors is fine-tuned.

2. **Diagonal vs. Random Elements:** We investigated the structural importance of the singular value matrix, $\Sigma_t$. We train randomly selected elements of $\Sigma_t$ (allowing for off-diagonal interactions).

In both cases, the proposed approach outperformed the alternatives as seen in Table 5. These results confirm that recalibrating the importance of stable basis vectors via the principal singular values is superior to directly modifying the vectors or introducing arbitrary off-diagonal terms.

### A.5   INVESTIGATING TRANSFER BOUNDARIES

While our primary objective was the direct comparison against the aTLAS baseline, necessitating a shared initialization for our main evaluation, we designed an entirely new experiments to test the practical boundaries of the AXIS methodology.

The study consisted of four distinct experimental runs, three of them utilizing a task vector derived from the fine-tuning on the Cars dataset:

    **A. Baseline:** We measured the fine-tuning performance of a randomly selected ViT-B/16 Cars task vector on all target tasks, using the standard ViT-B/16 base model. This served as our internal compatibility benchmark.

    **B. Cross-Initialization Transfer (Minor Architectural Changes):** We tested transfer between two distinct base models (ViT-B/32 and ViT-B/16) that share the same pre-training objective (CLIP) but represent different initializations and minor architectural differences (patch size). The task vector was computed as the difference between the fine-tuned and base parameters for ViT-B/32: $\Delta_{\text{task}} = \theta_{\text{ViT-B-32, Cars}} - \theta_{\text{ViT-B-32, base}}$. We applied our full AXIS adaptation methodology to the ViT-B/16 base model, explicitly skipping the two layers with shape mismatches while transferring knowledge from all other compatible layers. This configuration simultaneously tests cross-initialization transfer and adaptability to minor architectural differences.

    **C. Different Pre-training (Same Source Task):** To test the necessity of a shared initialization history, we utilized an architecturally compatible model with an entirely different pre-training source. We took the OpenClip ViT-B/16 model and fine-tuned it on the Cars dataset using the standard aTLAS recipe to create a new source task vector. The task vector was calculated as $\Delta_{\text{task}} = \theta_{\text{OpenClip, Cars}} - \theta_{\text{OpenClip, base}}$, and then applied to the CLIP ViT-B/16 base model without altering the AXIS procedure.

    **D. Out-of-Distribution Control:** As a control, we employed Microsoft's popular Biomed-CLIP (ViT-B/16) from HuggingFace. This domain-specific model, fine-tuned on medical images, serves as a highly distant reference point. The model is architecturally compatible but pre-trained and fine-tuned on data completely unrelated to our broad category of target tasks. We computed the task vector as the difference between the fine-tuned and base parameters of OpenClip (similar to C) and applied it to the CLIP ViT-B/16 base model.

The results for $N$=0.1 are presented in Table 6, which distinguishes the impact of the source task vector across the tested boundaries.

Overall, the results demonstrate that AXIS successfully achieves knowledge transfer despite differences in initialization and minor architectural changes. The hierarchy of transfer compatibility is clear: the highest performance benchmark remains the most compatible version (the native ViT-B/16 baseline), followed closely by the ViT-B/32 task vector and the OpenClip task vector. This shows a consistent relationship between base model compatibility and final performance, with both cross-initialization and minor architectural changes only leading to a slight decrease in average accuracy (from $80.59\%$ to $79.41\%$ and $79.16\%$, respectively).

The performance of the domain-distant BiomedCLIP source, while low on average, confirmed its role as an extreme sanity check. Crucially, even this highly specialized model provided positive transfer, exceeding the zero-shot capabilities of the ViT-B/16 base model on certain target tasks (e.g., EuroSAT). This result, expected given the model's highly specific domain knowledge, confirms that our method could function robustly even when provided with low-relevance source model.

The decision to utilize only a single source task vector for these boundary tests was deliberate, allowing us to avoid the methodological ambiguity that incorporating many other compatible source task vectors might mask poor performance.

| Dataset | A. ViT-B/16 | B. ViT-B/32 | C. OpenCLIP ViT-B/16 | D. BiomedCLIP |
|---|---|---|---|---|
| CIFAR10 | 97.17% | 97.30% | 97.11% | 33.84% |
| CIFAR100 | 80.73% | 80.91% | 80.37% | 5.71% |
| CUB200 | 63.43% | 61.93% | 61.13% | 0.90% |
| Caltech101 | 96.14% | 95.62% | 94.87% | 17.45% |
| Caltech256 | 91.49% | 91.39% | 90.85% | 5.23% |
| Country211 | 23.89% | 24.11% | 23.40% | 1.09% |
| DTD | 59.84% | 54.15% | 54.63% | 4.41% |
| EuroSAT | 97.56% | 97.19% | 97.19% | 51.30% |
| FGVCAircraft | 34.05% | 30.42% | 30.78% | 1.35% |
| Flowers102 | 78.13% | 74.37% | 74.09% | 0.47% |
| Food101 | 91.01% | 91.16% | 90.96% | 3.80% |
| GTSRB | 90.91% | 88.33% | 87.93% | 14.45% |
| MNIST | 98.33% | 97.98% | 97.95% | 20.60% |
| OxfordIIITPet | 93.40% | 93.49% | 93.21% | 3.82% |
| PascalVOC | 88.42% | 88.17% | 87.87% | 36.79% |
| RESISC45 | 89.46% | 88.22% | 88.65% | 14.30% |
| STL10 | 99.02% | 99.12% | 99.08% | 27.29% |
| SUN397 | 71.26% | 70.02% | 70.26% | 0.63% |
| SVHN | 90.56% | 89.49% | 90.13% | 24.51% |
| UCF101 | 76.92% | 74.83% | 72.80% | 4.47% |
| **Average** | **80.59%** | **79.41%** | **79.16%** | **13.62%** |

Table 6: Transfer performance of the different task vectors under varying initialization and architecture constraints ($N$=0.1).

## A.6 TRANSFERABILITY BY SVD COMPONENTS

### A.6.1 INDIVIDUAL COMPONENT ANALYSIS

In this experiment, we incrementally tested the performance of single components. Specifically, during Stage 2, we fine-tuned only one singular value per 2D layer in $\Delta_{target}$. This allowed us to evaluate which structural components contribute most effectively to the target task.

The averaged performance across all target tasks is illustrated in Figure 11. The results demonstrate a clear trend: higher raw singular values correlate with a significantly better capacity to transfer knowledge. As observed in the figure, the most structurally dominant component achieves an accuracy of approximately 61.6%, whereas performance drops sharply for lower-ranked components (stabilizing between 30% and 35%).

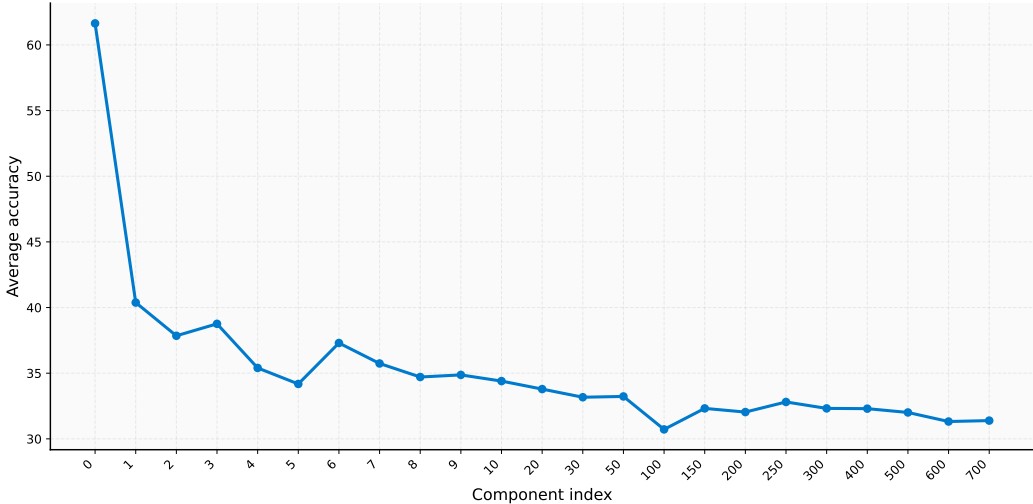

Figure 11: **Impact of singular value rank on transfer performance.** The plot illustrates the average accuracy obtained when fine-tuning individual singular components isolated by rank. The sharp decline demonstrates that the most structurally dominant components encapsulate the majority of transferable knowledge.

### A.6.2 GROUND-TRUTH ALIGNMENT ANALYSIS

Our objective is to empirically demonstrate that the singular vectors corresponding to the largest singular values in source tasks are, in fact, the most transferable to a new, unseen target task. We show that these top component directions align most closely with the ground-truth update required for the target task, independent of their original scalar singular values.

To quantify the transferability of specific components, we conduct an analysis where we assume access to the ground-truth target task vector, denoted as $\Delta_{target}$. This allows us to measure how well specific components from source tasks can reconstruct or "explain" the target task update.

We assume access to the ground-truth target task vector $\Delta_{target}$. For each source task $i$, we decompose its task matrix via SVD:

$$\Delta_{source}^{(i)} = \sum_{k=1}^{T} \sigma_k^{(i)} \mathbf{u}_k^{(i)} (\mathbf{v}_k^{(i)})^\top, \tag{1}$$

where the singular values $\sigma_k^{(i)}$ are sorted in descending order and $\mathbf{u}_k^{(i)}, \mathbf{v}_k^{(i)}$ are the left and right singular vectors. We evaluate for each layer $l$ the transferability of each rank-1 component $(\mathbf{u}_k^{(i)}, \mathbf{v}_k^{(i)})$ using the Preserved Energy metric (ignoring $\sigma_k^{(i)}$):

$$E(k, i) = \frac{\langle \Delta_{target}, \mathbf{u}_k^{(i)} (\mathbf{v}_k^{(i)})^\top \rangle^2}{\|\Delta_{target}\|_F^2}, \tag{2}$$

which measures the fraction of target variance explained by the component. To visualise the relationship between component rank $k$ and transferability, we aggregate $E(k, i)$ across all layers and tasks, ordering components by their original rank (all $k = 1$ components first, then $k = 2$, etc.), allowing comparison of average transferability of top versus bottom components globally.

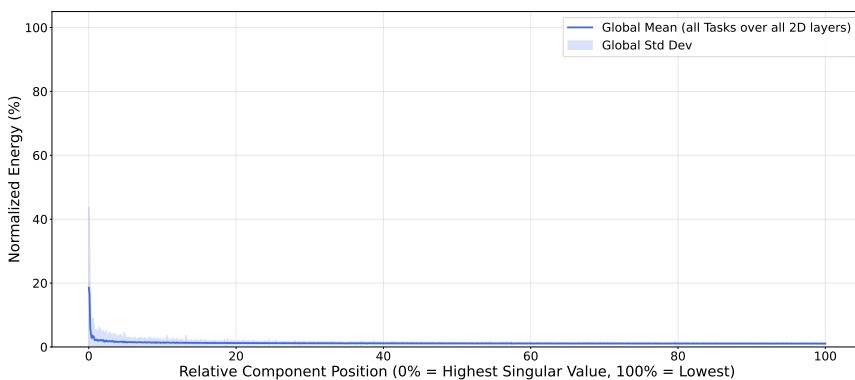

Figure 12: Normalized Energy Distribution (Preserved Energy) as a function of component rank. The x-axis represents the component index, sorted by original singular value magnitude (descending), while the y-axis shows the projection of source components onto the ground-truth target update $\Delta_{target}$. The displayed values are averaged across all source models and unseen target tasks for every layer matrix. The plot empirically confirms that components with the highest singular values in source tasks exhibit the highest transferability to the target task.

The resulting plot, Figure 12, displays the Normalized Energy Distribution. The x-axis represents the component index sorted by their original singular value rank (descending).

- **Peak Energy at Low Ranks:** The plot reveals a distinct concentration of high preserved energy values at the very beginning of the x-axis. These correspond to the components with the highest singular values ($k = 1, 2, \dots$) in their respective source tasks.
- **Rapid Decay:** As we move to higher indices (corresponding to lower singular values in source tasks), the preserved energy drops and plateaus.
- **Consistency:** This pattern holds true when averaged across all target tasks and layers, indicating a universal property of the task vector space.

The experiment demonstrates a strong empirical link between singular value magnitude and cross-task transferability. The components that are dominant (have large singular values) in the source tasks are consistently the ones that align best with the ground-truth direction of the target task.

### A.7 AXIS PERFORMANCE AT FULL PARAMETER BUDGET

The full-parameter performance of AXIS was evaluated by setting the singular value budget to $N$=100%. The method achieved an average accuracy of 82.30%, which closely approaches the standard full fine-tuning baseline 83.56%. Crucially, this near-equivalent performance was attained using a fixed, significantly shorter training schedule of 10 epochs across all datasets, in contrast to the baseline's optimized, dataset-specific training that required up to 76 epochs. AXIS approaches the performance of full fine-tuning while maintaining superior computational efficiency.

### A.8 MULTI-TASK PERFORMANCE OF AXIS

In scenarios involving high-throughput streams of new tasks, restarting the adaptation process for every individual target task may become a computational bottleneck. To evaluate the training efficiency of our framework, we implemented a joint multi-task adaptation strategy. Instead of performing independent fine-tuning runs for each target dataset, we conducted a single training session where the AXIS model was adapted simultaneously on a combined dataset comprising six distinct target tasks (Flowers102, OxfordIIITPet, CUB200, PascalVOC, Country211, UCF101). In this setup, the learnable singular values ($\Sigma_t$) were shared across all tasks, while task-specific classification heads were maintained to handle distinct label spaces. The comparison between this joint approach and the standard individual approach is illustrated in Figure 13. We conducted analysis using the ViT-B-32

| Dataset Name | Fully Finetuned | Axis |
|---|---|---|
| CIFAR10 | 98.05 | 97.67 |
| CIFAR100 | 89.09 | 84.78 |
| CUB200 | 73.56 | 66.30 |
| Caltech101 | 76.41 | 94.89 |
| Caltech256 | 92.60 | 90.64 |
| Cars | 78.26 | 72.33 |
| Country211 | 21.99 | 19.79 |
| DTD | 78.94 | 74.49 |
| EuroSAT | 98.89 | 98.65 |
| FGVCAircraft | 40.65 | 44.74 |
| Flowers102 | 90.08 | 85.92 |
| Food101 | 94.41 | 87.88 |
| GTSRB | 99.14 | 95.92 |
| MNIST | 99.65 | 99.10 |
| OxfordIIITPet | 92.15 | 90.22 |
| PascalVOC | 88.42 | 86.79 |
| RESISC45 | 95.94 | 93.53 |
| STL10 | 88.68 | 97.09 |
| SUN397 | 75.40 | 71.86 |
| SVHN | 97.38 | 94.67 |
| UCF101 | 85.01 | 81.01 |
| Average | 83.56 | 82.30 |

Table 7: Comparison of AXIS with $N$=100% with the full-parameters finetuning in ViT-B-32 architecture.

architecture with three seeds across three trainable parameter budgets ($N \in \{10\%, 20\%, 40\%\}$). To ensure a fair comparison, the total computational budget was equalized between the two strategies. Specifically, the joint multi-task model was trained for an extended number of epochs, equivalent to the cumulative training steps of the six individual adaptations.

The standard individual adaptation strategy outperforms the joint approach across all parameter budgets. Current AXIS design is optimized for high-fidelity, task-specific specialization. Effective application in a simultaneous multi-task setting require further modifications to mitigate interference between conflicting task gradients.

## A.9 ZERO-SHOT TRANSFERABILITY OF TRAINED MODELS

To evaluate the specificity of the learned adaptations, we assessed the cross-task performance of AXIS models. We selected a subset of six distinct target tasks and trained individual models using the ViT-B-32 architecture with a parameter budget of $N = 10\%$. Each adapted model was then evaluated on all other target datasets in a zero-shot manner. As illustrated in the heatmap in Figure 14, the models exhibit strong task orientation: while they achieve high accuracy on their respective target datasets, their performance on unseen tasks is significantly lower. This confirms that the task-specific specialization achieved in Stage 2 comes at the natural cost of zero-shot transferability.

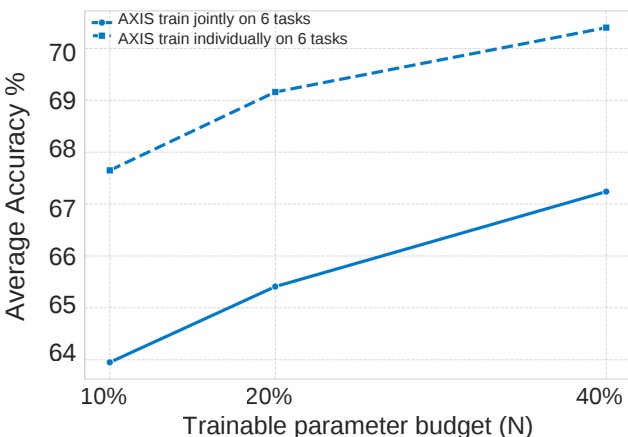

Figure 13: Comparison of average accuracy across six target tasks between the joint multi-task adaptation strategy and the standard individual adaptation, demonstrating that task-specific fine-tuning yields superior performance compared to simultaneous joint training across all trainable parameter budgets

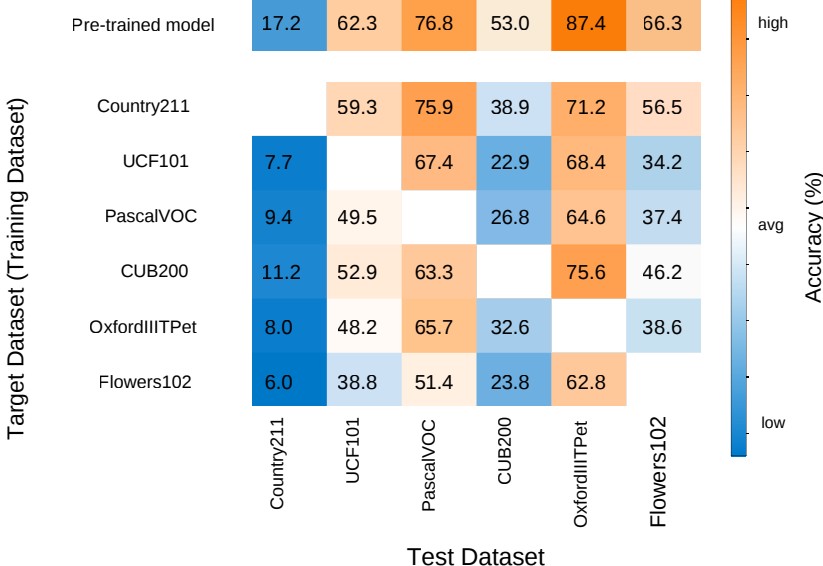

Figure 14: Cross-task transferability heatmap (ViT-B-32, $N = 10\%$) demonstrating that AXIS models exhibiting limited zero-shot generalization to unseen datasets.

## A.10 DYNAMIC TOP-K SELECTION

We compare fixed top-$K$ selection with the Optimal Hard Thresholding Donoho & Gavish (2013) method. This method automatically determines the optimal cut-off for singular values based on the estimated noise level of the matrix. We compared the performance of AXIS using this automatic thresholding against our fixed strategy on the ViT-B-32 architecture for each task matrix. The automatic method achieved an average accuracy of 64.5%, compare to 78.42% achieved by AXIS. Given that it does not provide a tangible performance gain, we retain the fixed strategy as the preferred approach for its balance of simplicity, performance, and computational predictability.

## A.11 INCREMENTAL KNOWLEDGE AGGREGATION

To evaluate the adaptability of AXIS to dynamic scenarios where source models arrive sequentially, we examine an *incremental aggregation protocol*. While the default framework performs a global ranking over the singular components of the entire pool of source task matrices $\{\Delta_1, \ldots, \Delta_{T-1}\}$, the incremental variant updates the merged knowledge base iteratively.

Specifically, we initialize the merged matrix $\Delta_m$ with the first two source task vectors. For each subsequent incoming source $\Delta_i$, we treat the currently accumulated matrix $\Delta_m^{(i-1)}$ as a consolidated representation of prior knowledge and merge it with the new source. This recursive update rule allows us to apply the aggregation mechanism defined in Stage 1 pairwise:

$$\Delta_m^{(i)} = \text{Stage1}(\{\Delta_m^{(i-1)}, \Delta_i\}) \tag{3}$$

In this setup, the aggregation step selects the top-$K$ components from the union of the accumulated basis and the new task vector. Crucially, this creates a memory-efficient online process where historical source parameters are discarded immediately after integration, eliminating the need for a persistent buffer of previous models.

With an average accuracy of 78.55%, the streaming approach performs on par with the standard global ranking protocol (78.48%). Consequently, AXIS proves effective at real-time structural knowledge accumulation, eliminating the overhead associated with iteratively processing the full model history.

## A.12 IN-DEPTH ROBUSTNESS ANALYSES

### A.12.1 ROBUSTNESS TO INPUT PERTURBATIONS

To further probe the robustness capabilities of AXIS and aTLAS, we evaluate them against a set of 12 common image corruptions Hendrycks & Dietterich (2019). Each corruption type is applied to the test set of target task images at five distinct severity levels to simulate a range of degradations. As illustrated in Figure 16, our proposed method, AXIS, maintains a slightly average performance advantage (0.83 percentage points). This margin is particularly pronounced for corruption types where overall accuracy remains high, indicating better robustness in moderately challenging conditions. A detailed breakdown by severity level delineates this trend more clearly (see Figure 17). AXIS demonstrates greater resilience across the initial four perturbation levels, outperforming aTLAS by margins of 2.04 percentage points for the lowest corruption severity.

Furthermore, we extend our robustness evaluation to scenarios with partial input information, a challenge simulated using patch dropout. A detailed, step-by-step analysis, presented in Table 8, illustrates how the model's resilience to input masking evolves as the incremental aggregation of each source task vector is performed. This granular breakdown demonstrates that the fusion of diverse knowledge sources enhances the model's ability to perform predictions even when significant portions of the input are omitted.

### A.12.2 TRAINING DATA AVAILABILITY

To assess the data efficiency of our approach and its robustness in limited data scenarios, we investigate the performance of our method compared to aTLAS under varying levels of training data availability for the target task. For this experiment, we reduce the size of the target task's training dataset, creating subsets with 5%, 10%, 25%, 50%, 75%, and 95% of the original samples. The results, illustrated in Figure 18, demonstrate that our method maintains a significant performance advantage over aTLAS across the broad majority of data availability levels.

### A.12.3 ROBUSTNESS AGAINST ALTERED SOURCE PARAMETERS

For a detailed analysis of the framework's robustness, we refer to Table 19 and Figure 20, which provides a comprehensive performance breakdown under two challenging scenarios: contamination by a single noisy source vector and aggregation of heavily pruned (95%) source vectors.

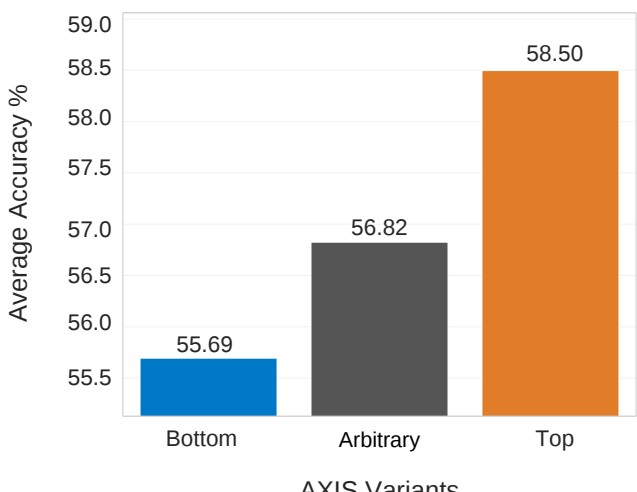

Figure 15: AXIS with top-component selection is more robust against common corruptions (e.g., blur) than other strategies.

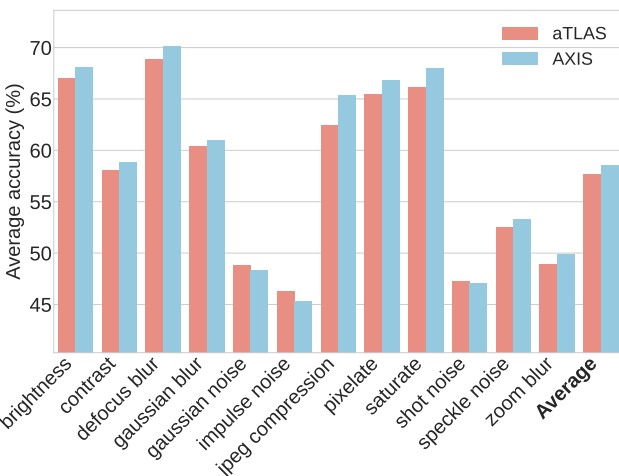

Figure 16: The accuracy across each type of corruption is evaluated for all severity levels ranging from 1 to 5 for all 21 target tasks.

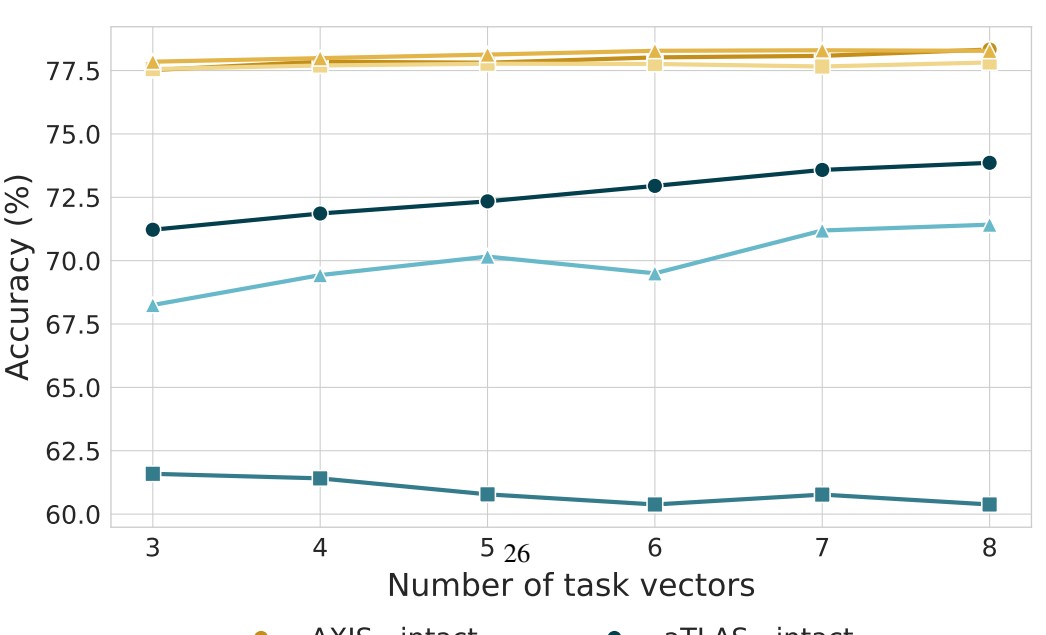

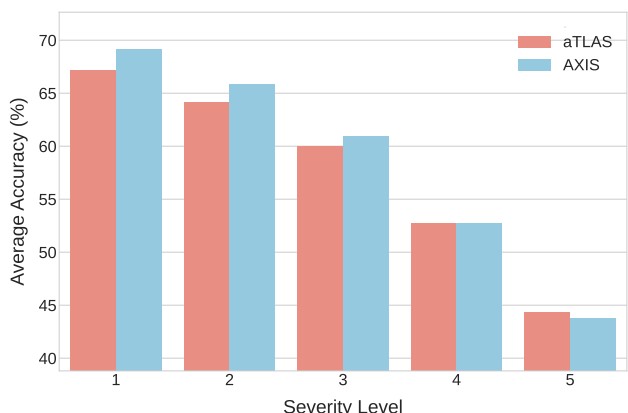

Figure 17: Severity levels average over all 12 image corruptions.

| TV | | | | Input Patch Dropout (%) | | | | | |
|---|---|---|---|---|---|---|---|---|---|
| | 77.31 | 75.53 | 72.78 | 67.51 | 63.61 | 56.86 | 52.11 | 44.66 | 39.79 | 28.74 |
| 1 | | | | | | | | | | |
| 2 | 77.51 (+0.20) | 75.80 (+0.27) | 73.40 (+0.62) | 69.04 (+1.53) | 65.38 (+1.77) | 59.54 (+2.68) | 55.17 (+3.05) | 48.36 (+3.69) | 43.63 (+3.84) | 32.67 (+3.93) |
| 3 | 77.52 (+0.21) | 75.88 (+0.35) | 73.49 (+0.71) | 69.27 (+1.76) | 65.72 (+2.12) | 60.00 (+3.13) | 55.87 (+3.75) | 49.28 (+4.62) | 44.84 (+5.05) | 34.08 (+5.34) |
| 4 | 77.85 (+0.54) | 76.12 (+0.60) | 73.68 (+0.89) | 69.43 (+1.91) | 65.92 (+2.31) | 59.97 (+3.10) | 55.74 (+3.62) | 49.12 (+4.45) | 44.58 (+4.79) | 33.49 (+4.75) |
| 5 | 77.81 (+0.50) | 76.24 (+0.71) | 73.98 (+1.20) | 69.82 (+2.30) | 66.51 (+2.90) | 60.95 (+4.08) | 56.87 (+4.76) | 50.27 (+5.60) | 45.65 (+5.86) | 34.59 (+5.86) |
| 6 | 78.02 (+0.71) | 76.41 (+0.88) | 73.96 (+1.18) | 69.78 (+2.27) | 66.40 (+2.80) | 60.53 (+3.66) | 56.54 (+4.42) | 50.10 (+5.43) | 45.63 (+5.84) | 34.64 (+5.90) |
| 7 | 78.08 (+0.76) | 76.48 (+0.95) | 74.23 (+1.45) | 70.00 (+2.49) | 66.82 (+3.21) | 61.16 (+4.30) | 57.01 (+4.90) | 50.43 (+5.77) | 45.74 (+5.95) | 34.45 (+5.71) |
| 8 | 78.33 (+1.02) | 76.67 (+1.14) | 74.27 (+1.49) | 70.27 (+2.76) | 66.96 (+3.35) | 61.21 (+4.35) | 57.13 (+5.01) | 50.71 (+6.04) | 46.12 (+6.33) | 34.98 (+6.24) |
| 9 | 78.41 (+1.10) | 76.74 (+1.21) | 74.42 (+1.64) | 70.29 (+2.78) | 66.88 (+3.27) | 61.49 (+4.63) | 57.63 (+5.52) | 51.24 (+6.57) | 47.02 (+7.23) | 36.10 (+7.36) |
| 10 | 78.16 (+0.85) | 76.60 (+1.07) | 74.20 (+1.42) | 69.85 (+2.33) | 66.37 (+2.77) | 60.55 (+3.69) | 56.32 (+4.20) | 49.77 (+5.11) | 45.17 (+5.38) | 34.25 (+5.51) |
| 11 | 78.40 (+1.09) | 76.87 (+1.34) | 74.44 (+1.66) | 70.29 (+2.78) | 66.81 (+3.20) | 60.74 (+3.88) | 56.21 (+4.09) | 49.26 (+4.59) | 44.47 (+4.68) | 32.81 (+4.07) |
| 12 | 78.51 (+1.20) | 76.90 (+1.38) | 74.56 (+1.78) | 70.34 (+2.83) | 66.92 (+3.32) | 61.11 (+4.24) | 57.05 (+4.93) | 50.31 (+5.65) | 45.78 (+5.99) | 33.99 (+5.26) |
| 13 | 78.37 (+1.06) | 76.71 (+1.19) | 74.34 (+1.55) | 70.15 (+2.63) | 66.78 (+3.17) | 61.02 (+4.15) | 56.85 (+4.74) | 49.97 (+5.31) | 45.32 (+5.53) | 33.81 (+5.08) |
| 14 | 78.41 (+1.10) | 76.83 (+1.30) | 74.42 (+1.64) | 70.16 (+2.65) | 66.75 (+3.15) | 60.87 (+4.00) | 56.71 (+4.60) | 49.65 (+4.98) | 44.90 (+5.10) | 33.05 (+4.31) |
| 15 | 78.34 (+1.02) | 76.81 (+1.28) | 74.50 (+1.71) | 70.28 (+2.77) | 66.82 (+3.21) | 60.74 (+3.87) | 56.32 (+4.20) | 49.24 (+4.57) | 44.64 (+4.85) | 33.14 (+4.40) |
| 16 | 78.42 (+1.11) | 76.85 (+1.32) | 74.70 (+1.92) | 70.45 (+2.94) | 67.11 (+3.50) | 61.37 (+4.50) | 56.96 (+4.85) | 50.16 (+5.50) | 45.73 (+5.94) | 34.51 (+5.78) |
| 17 | 78.41 (+1.09) | 76.82 (+1.29) | 74.57 (+1.79) | 70.38 (+2.87) | 67.06 (+3.45) | 61.32 (+4.45) | 57.10 (+4.98) | 50.41 (+5.74) | 45.93 (+6.14) | 34.91 (+6.17) |
| 18 | 78.54 (+1.23) | 76.94 (+1.41) | 74.63 (+1.85) | 70.53 (+3.01) | 67.36 (+3.76) | 61.77 (+4.91) | 57.62 (+5.51) | 50.92 (+6.26) | 46.45 (+6.66) | 34.92 (+6.19) |
| 19 | 78.58 (+1.27) | 76.91 (+1.38) | 74.61 (+1.83) | 70.20 (+2.69) | 66.87 (+3.26) | 61.19 (+4.32) | 56.97 (+4.86) | 50.48 (+5.82) | 46.05 (+6.26) | 34.64 (+5.90) |
| 20 | 78.50 (+1.19) | 76.75 (+1.22) | 74.51 (+1.73) | 70.14 (+2.63) | 66.93 (+3.32) | 61.19 (+4.33) | 57.25 (+5.13) | 50.58 (+5.91) | 46.31 (+6.52) | 35.05 (+6.31) |
| **21** | **78.48** (+1.16) | **76.82** (+1.29) | **74.68** (+1.90) | **70.52** (+3.00) | **67.37** (+3.76) | **61.84** (+4.97) | **57.82** (+5.71) | **51.18** (+6.51) | **46.63** (+6.83) | **35.39** (+6.65) |

Table 8: Performance analysis of AXIS under increasing input masking. The table illustrates that aggregating more source task vectors (TV) enhances model robustness to input patch dropout. We report the mean accuracy (%) across all target tasks for dropout rates from 0% to 50%. Each row corresponds to a different number of aggregated sources, and values in parentheses show the improvement in percentage points (p.p.) over the first, single task vector baseline (first row).

### A.13    Component Selection

To study our hypothesis that the most useful transferable knowledge is encapsulated within the principal singular components, we conducted a comprehensive ablation study. We evaluated the impact of different component selection and aggregation strategies on final model performance. The goal was to ensure that our default approach, aggregating components with the highest singular values, is effective to other plausible alternatives, especially with the highest number of source task vectors. We compared the following seven strategies:

- **Top Components (our default):** As described in the main paper, we perform a global ranking of all singular components from all source tasks and select the top-K based on their singular values ($\sigma_k$) to form the merged matrix $\Delta_m = \sum_{k=1}^{K} u_k \sigma_k v_k^\top$.
- **Bottom Components:** A control strategy where we select the K components with the lowest singular values from the global ranking.
- **Arbitrary Components:** A second control strategy where K components are arbitrarily selected from the global pool.
- **Average Top Components:** This baseline first distills each source task matrix $\Delta_i$ into its top-K principal components. Next, all these resulting low-rank matrices are averaged into a

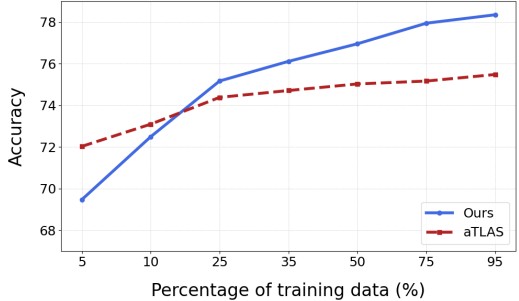

| Task | aTLAS | | | AXIS (ours) | | |
| Vectors | intact | corrupted | pruned | intact | corrupted | pruned |
|---|---|---|---|---|---|---|
| 3 | 71.22 | 61.59 | 68.25 | 77.52 | 77.56 | 77.85 |
| 4 | 71.86 | 61.41 | 69.43 | 77.85 | 77.70 | 77.99 |
| 5 | 72.34 | 60.78 | 70.16 | 77.81 | 77.77 | 78.13 |
| 6 | 72.95 | 60.38 | 69.50 | 78.02 | 77.76 | 78.28 |
| 7 | 73.58 | 60.77 | 71.19 | 78.08 | 77.66 | 78.30 |
| 8 | 73.86 | 60.38 | 71.42 | 78.33 | 77.82 | 78.28 |

Figure 19: Robustness to altered source task vectors. AXIS shows higher resilience to corruption and pruning compared to aTLAS.

Figure 18: AXIS performs better with smaller amounts of training data in almost all cases.

single matrix. Finally, we perform a new SVD on this averaged matrix and select its top-K components to form the final $\Delta_m$.

- **Average Bottom Components:** The inverse of the "average top components" baseline, used as a control. First, each source task matrix is reduced to a low-rank approximation using only its own bottom-K singular components. Second, these resulting low-rank matrices are averaged, and a final selection of the bottom-K components is performed via SVD on this single, averaged matrix.

- **Equal Top Contribution:** This strategy ensures a balanced representation from all source tasks. Instead of a global ranking, it selects an equal number of the top singular components from each individual source task. If the total budget is K components and there are $T - 1$ sources, we select the top $K/(T - 1)$ components from each task. These are then pooled and summed to form $\Delta_m$.

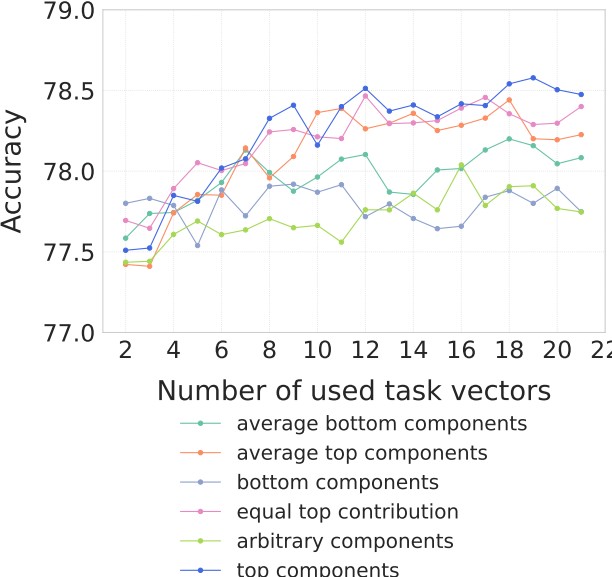

Figure 21: Performance comparison of seven different SVD component aggregation strategies $K$ with constant $N$=10%. The plot shows the average accuracy across all target tasks as the number of used source task vectors increases. Our default strategy, top components, yields the best performance with the largest number of sources.

The results, presented in Figure 21, demonstrate that the top components strategy slightly outperforms on average all other alternatives across a varying number of aggregated source tasks. For

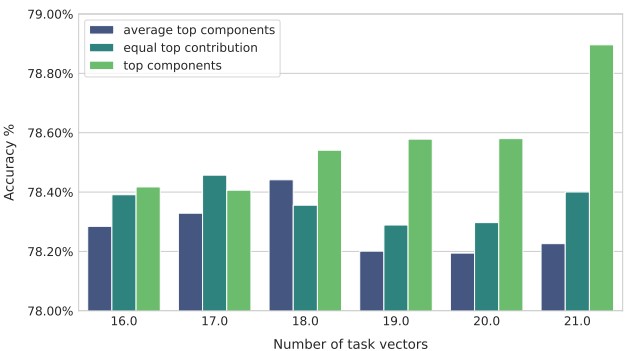

Figure 22: Detailed performance comparison of SVD component aggregation strategies, focusing on small variations within top components. While all strategies show comparable performance, the top components generally maintain a slight edge, particularly with a higher number of aggregated source tasks.

| Aggregated Task Vectors | Top | | Bottom | | Average top | | Average bottom | |
|---|---|---|---|---|---|---|---|---|
| | SVD ✓ | SVD ✗ | SVD ✓ | SVD ✗ | SVD ✓ | SVD ✗ | SVD ✓ | SVD ✗ |
| 1 | 77.31 | 77.35 | 77.63 | 77.57 | 77.42 | 77.42 | 77.62 | 77.25 |
| 2 | 77.51 | 77.38 | 77.80 | 77.65 | 77.42 | 77.41 | 77.58 | 77.23 |
| 3 | 77.52 | 76.36 | 77.83 | 77.80 | 77.41 | 77.37 | 77.74 | 77.29 |
| 4 | 77.85 | 76.49 | 77.79 | 77.61 | 77.74 | 77.75 | 77.74 | 77.33 |
| 5 | 77.81 | 76.56 | 77.54 | 77.75 | 77.86 | 77.83 | 77.82 | 77.14 |
| 6 | 78.02 | 76.39 | 77.88 | 77.85 | 77.85 | 77.95 | 77.93 | 77.35 |
| 7 | 78.08 | 76.40 | 77.72 | 77.85 | 78.14 | 78.20 | 78.13 | 77.51 |
| 8 | 78.33 | 71.16 | 77.91 | 77.84 | 77.96 | 77.98 | 77.99 | 77.53 |
| 9 | 78.41 | 69.85 | 77.92 | 77.85 | 78.09 | 78.13 | 77.88 | 77.64 |
| 10 | 78.16 | 71.58 | 77.87 | 77.84 | 78.36 | 78.24 | 77.96 | 77.50 |
| 11 | 78.40 | 78.52 | 77.92 | 77.84 | 78.39 | 78.42 | 78.07 | 77.49 |
| 12 | 78.51 | 78.39 | 77.72 | 77.80 | 78.26 | 78.28 | 78.10 | 77.34 |
| 13 | 78.37 | 78.49 | 77.80 | 77.77 | 78.30 | 78.38 | 77.87 | 77.68 |
| 14 | 78.41 | 78.37 | 77.71 | 77.72 | 78.36 | 78.20 | 77.86 | 77.54 |
| 15 | 78.34 | 78.53 | 77.64 | 77.66 | 78.25 | 78.21 | 78.01 | 77.52 |
| 16 | 78.42 | 78.57 | 77.66 | 77.76 | 78.28 | 78.29 | 78.02 | 77.36 |
| 17 | 78.41 | 78.51 | 77.84 | 77.77 | 78.33 | 78.28 | 78.13 | 77.56 |
| 18 | 78.54 | 78.60 | 77.88 | 77.76 | 78.44 | 78.30 | 78.20 | 77.70 |
| 19 | 78.58 | 78.50 | 77.80 | 77.65 | 78.20 | 78.31 | 78.16 | 77.55 |
| 20 | 78.50 | 78.45 | 77.89 | 77.79 | 78.19 | 78.18 | 78.05 | 77.64 |
| 21 | 78.48 | 78.49 | 77.75 | 77.78 | 78.23 | 78.27 | 78.08 | 77.37 |

Table 9: Performance comparison of different aggregation strategies with and without the final SVD step, across a varying number of aggregated task vectors and different component selection strategies.

example, the top components strategy achieved an average score of 78.23 across all used task vectors, slightly edging out the equal top contribution approach, which averaged 78.19.

Additionally, we compare how different selection strategies for the top-ranking components affect accuracy when using the largest number of source task vectors, as illustrated in Figure 22. For this configuration, the top components strategy yielded the highest accuracy. These results are averaged across all target tasks. Additionally, we provided detailed results on the main aggregation strategies per target dataset in the Table 13.

## A.14 IMPACT OF FINAL SVD

To empirically validate the importance of the final SVD re-parameterization, as discussed in the main text, we conduct a detailed ablation study. Table 9 presents a performance comparison of four different component aggregation strategies, each evaluated with and without the final SVD step.

The omission of the final SVD step (denoted as 'SVD X') is particularly detrimental to the top components strategy, resulting in a significant performance drop (e.g., over eight percentage points when aggregating 9 task vectors). In contrast, strategies based on bottom or average components exhibit significantly higher resilience to this omission. We hypothesize that two related factors drive this phenomenon. First, the top components, representing high-magnitude task-specific knowledge, likely exhibit more substantial destructive interference when their non-orthogonal vectors are directly summed. Second, this instability may be amplified during the fine-tuning process. Without a shared orthogonal basis provided by the final SVD, the learnable parameters (a subset of singular values) may conflict with the frozen components, as their underlying vectors are not decorrelated. This could lead to an unstable optimization process where adjustments to learnable components negatively interfere with the knowledge stored in the frozen ones. The relative stability of the bottom components strategy suggests that the interference from low-magnitude components is negligible, making the final orthogonalization beneficial but not as critical.

## A.15 DETAILED MAIN RESULTS

For a comprehensive and granular evaluation of our proposed framework, Tables 10–12 present a detailed, per-dataset comparison of AXIS and the aTLAS baseline.

| STV | Method | CIFAR100 | CIFAR10 | CUB200 | Caltech101 | Caltech256 | Cars | Country211 |
|-----|--------|----------|---------|--------|------------|------------|------|------------|
| 1 | aTLAS (N=10%) | 72.95 | 93.76 | 54.47 | 89.86 | 85.10 | 61.21 | 17.69 |
| | aTLAS (N=20%) | 73.62 | 94.15 | 55.38 | 91.65 | 85.53 | 62.12 | 17.92 |
| | aTLAS (N=40%) | 75.09 | 95.20 | 56.80 | 93.38 | 87.59 | 63.77 | 18.05 |
| | AXIS (N=10%) | 77.00 | 95.85 | 57.61 | 93.89 | 88.44 | 63.54 | 17.70 |
| | AXIS (N=20%) | 79.28 | 96.63 | 60.15 | 94.41 | 89.19 | 65.58 | 18.39 |
| | AXIS (N=40%) | 81.45 | 97.10 | 62.50 | 94.99 | 89.38 | 65.94 | 18.64 |
| 5 | aTLAS (N=10%) | 73.90 | 94.52 | 54.83 | 91.53 | 85.43 | 62.06 | 17.78 |
| | aTLAS (N=20%) | 74.77 | 95.17 | 55.94 | 92.68 | 87.59 | 62.53 | 18.02 |
| | aTLAS (N=40%) | 75.29 | 95.31 | 56.85 | 93.78 | 88.06 | 63.89 | 18.17 |
| | AXIS (N=10%) | 77.51 | 96.50 | 58.41 | 93.61 | 88.01 | 63.95 | 18.17 |
| | AXIS (N=20%) | 79.96 | 96.84 | 59.22 | 94.70 | 89.48 | 67.23 | 18.60 |
| | AXIS (N=40%) | 82.28 | 97.13 | 62.46 | 94.24 | 89.89 | 69.69 | 18.86 |
| 10 | aTLAS (N=10%) | 78.92 | 96.40 | 55.11 | 91.88 | 86.21 | 62.37 | 18.06 |
| | aTLAS (N=20%) | 79.68 | 96.58 | 55.78 | 93.72 | 86.82 | 62.90 | 18.24 |
| | aTLAS (N=40%) | 80.65 | 96.90 | 55.47 | 94.82 | 88.29 | 64.15 | 18.41 |
| | AXIS (N=10%) | 80.09 | 96.96 | 57.85 | 94.82 | 88.76 | 64.66 | 18.24 |
| | AXIS (N=20%) | 81.31 | 97.10 | 59.58 | 94.82 | 89.53 | 67.07 | 18.08 |
| | AXIS (N=40%) | 82.64 | 97.49 | 61.74 | 94.64 | 89.20 | 69.92 | 19.22 |
| 15 | aTLAS (N=10%) | 78.95 | 96.46 | 55.89 | 94.70 | 88.11 | 62.04 | 18.16 |
| | aTLAS (N=20%) | 79.81 | 96.81 | 57.08 | 95.22 | 89.19 | 64.15 | 18.30 |
| | aTLAS (N=40%) | 80.62 | 97.19 | 57.82 | 96.08 | 89.38 | 64.88 | 18.51 |
| | AXIS (N=10%) | 80.14 | 96.85 | 58.68 | 94.64 | 88.65 | 65.43 | 18.31 |
| | AXIS (N=20%) | 81.55 | 97.25 | 60.94 | 95.56 | 89.89 | 66.86 | 18.48 |
| | AXIS (N=40%) | 82.83 | 97.38 | 63.00 | 95.28 | 90.33 | 69.99 | 19.24 |
| 21 | aTLAS (N=10%) | 78.91 | 96.53 | 55.85 | 94.53 | 88.81 | 63.29 | 18.07 |
| | aTLAS (N=20%) | 79.94 | 96.79 | 57.46 | 94.64 | 89.43 | 64.21 | 18.36 |
| | aTLAS (N=40%) | 80.84 | 97.14 | 58.01 | 95.28 | 89.89 | 65.09 | 18.32 |
| | AXIS (N=10%) | 80.11 | 96.93 | 58.46 | 94.99 | 88.76 | 65.09 | 18.48 |
| | AXIS (N=20%) | 81.69 | 97.13 | 61.10 | 94.64 | 89.95 | 66.88 | 18.58 |
| | AXIS (N=40%) | 82.96 | 97.39 | 62.70 | 95.45 | 90.75 | 70.77 | 19.42 |

Table 10: Detailed results per target dataset for various numbers of source task vectors (STV). Part 1 of 3.

| STV | Method | DTD | EuroSAT | FGVCAircraft | Flowers102 | Food101 | GTSRB | MNIST |
|---|---|---|---|---|---|---|---|---|
| 1 | aTLAS (N=10%) | 48.78 | 88.81 | 22.62 | 67.39 | 85.11 | 54.90 | 82.44 |
| | aTLAS (N=20%) | 51.49 | 90.85 | 23.64 | 67.96 | 85.09 | 59.20 | 84.84 |
| | aTLAS (N=40%) | 56.97 | 95.04 | 24.75 | 70.25 | 85.73 | 78.45 | 93.38 |
| | AXIS (N=10%) | 67.02 | 97.30 | 29.70 | 77.49 | 85.81 | 89.57 | 97.36 |
| | AXIS (N=20%) | 70.80 | 97.70 | 30.66 | 81.15 | 86.28 | 93.20 | 98.46 |
| | AXIS (N=40%) | 74.15 | 98.30 | 19.65 | 81.20 | 86.93 | 94.22 | 98.76 |
| 5 | aTLAS (N=10%) | 53.03 | 94.11 | 22.86 | 68.56 | 85.27 | 66.85 | 89.08 |
| | aTLAS (N=20%) | 54.04 | 94.48 | 24.15 | 68.26 | 85.41 | 71.35 | 91.97 |
| | aTLAS (N=40%) | 58.67 | 95.44 | 24.83 | 69.58 | 85.86 | 79.96 | 93.44 |
| | AXIS (N=10%) | 65.69 | 97.41 | 30.48 | 77.22 | 86.05 | 90.74 | 97.78 |
| | AXIS (N=20%) | 70.96 | 97.63 | 33.75 | 80.09 | 86.62 | 93.45 | 98.57 |
| | AXIS (N=40%) | 73.09 | 98.22 | 16.83 | 82.84 | 87.02 | 94.51 | 98.81 |
| 10 | aTLAS (N=10%) | 55.96 | 95.59 | 24.18 | 69.02 | 85.27 | 77.00 | 95.42 |
| | aTLAS (N=20%) | 59.57 | 95.93 | 24.54 | 69.60 | 85.71 | 83.70 | 96.44 |
| | aTLAS (N=40%) | 64.26 | 96.93 | 26.70 | 72.30 | 85.94 | 88.06 | 97.25 |
| | AXIS (N=10%) | 68.14 | 98.00 | 31.95 | 76.65 | 86.15 | 90.02 | 98.02 |
| | AXIS (N=20%) | 70.85 | 98.33 | 29.70 | 79.10 | 86.49 | 93.61 | 98.54 |
| | AXIS (N=40%) | 71.91 | 98.19 | 19.20 | 77.82 | 87.07 | 94.73 | 98.96 |
| 15 | aTLAS (N=10%) | 56.44 | 95.15 | 24.93 | 70.22 | 85.60 | 78.31 | 96.15 |
| | aTLAS (N=20%) | 60.21 | 96.11 | 25.86 | 73.61 | 85.99 | 83.08 | 96.94 |
| | aTLAS (N=40%) | 62.71 | 96.81 | 28.14 | 74.48 | 86.17 | 87.39 | 97.06 |
| | AXIS (N=10%) | 67.82 | 97.78 | 31.05 | 77.25 | 86.18 | 91.00 | 98.20 |
| | AXIS (N=20%) | 70.59 | 98.19 | 34.92 | 82.09 | 86.61 | 93.67 | 98.70 |
| | AXIS (N=40%) | 71.38 | 98.26 | 39.15 | 83.67 | 87.11 | 94.76 | 98.89 |
| 21 | aTLAS (N=10%) | 56.44 | 95.07 | 25.62 | 71.23 | 85.72 | 78.02 | 95.98 |
| | aTLAS (N=20%) | 60.37 | 96.26 | 26.37 | 72.09 | 85.91 | 83.45 | 96.94 |
| | aTLAS (N=40%) | 63.24 | 96.96 | 26.25 | 75.09 | 86.29 | 88.38 | 97.58 |
| | AXIS (N=10%) | 67.98 | 97.81 | 30.75 | 77.87 | 86.32 | 91.06 | 98.11 |
| | AXIS (N=20%) | 70.64 | 98.22 | 34.50 | 82.31 | 86.57 | 93.46 | 98.64 |
| | AXIS (N=40%) | 72.18 | 98.52 | 38.97 | 83.74 | 87.15 | 94.43 | 98.96 |

Table 11: Detailed results per target dataset for various numbers of source task vectors (STV). Part 2 of 3.

| STV | Method | OxfordIIITPet | PascalVOC | RESISC45 | STL10 | SUN397 | SVHN | UCF101 |
|---|---|---|---|---|---|---|---|---|
| 1 | aTLAS (N=10%) | 90.19 | 82.99 | 71.19 | 97.99 | 64.42 | 62.10 | 65.05 |
| | aTLAS (N=20%) | 90.73 | 84.21 | 72.14 | 98.16 | 64.95 | 67.11 | 65.61 |
| | aTLAS (N=40%) | 90.71 | 86.51 | 80.40 | 98.49 | 66.16 | 86.49 | 68.94 |
| | AXIS (N=10%) | 89.92 | 85.77 | 87.51 | 97.65 | 66.80 | 86.63 | 71.00 |
| | AXIS (N=20%) | 89.86 | 86.77 | 89.95 | 97.80 | 68.48 | 89.76 | 74.91 |
| | AXIS (N=40%) | 90.24 | 86.53 | 91.84 | 97.08 | 70.05 | 91.35 | 77.98 |
| 5 | aTLAS (N=10%) | 90.62 | 85.49 | 74.56 | 97.91 | 64.85 | 83.23 | 65.95 |
| | aTLAS (N=20%) | 91.31 | 86.16 | 77.16 | 98.35 | 65.43 | 84.09 | 67.94 |
| | aTLAS (N=40%) | 91.99 | 86.72 | 80.43 | 98.34 | 66.29 | 86.34 | 68.86 |
| | AXIS (N=10%) | 90.60 | 86.71 | 87.90 | 97.74 | 67.27 | 90.87 | 71.45 |
| | AXIS (N=20%) | 90.19 | 87.09 | 90.41 | 97.65 | 68.69 | 92.18 | 76.37 |
| | AXIS (N=40%) | 90.27 | 86.99 | 91.90 | 97.26 | 69.75 | 92.87 | 78.03 |
| 10 | aTLAS (N=10%) | 91.77 | 86.19 | 79.13 | 98.24 | 66.33 | 85.66 | 67.57 |
| | aTLAS (N=20%) | 91.61 | 86.63 | 82.16 | 98.21 | 66.76 | 87.45 | 69.36 |
| | aTLAS (N=40%) | 91.50 | 87.11 | 84.87 | 98.24 | 67.04 | 89.06 | 71.66 |
| | AXIS (N=10%) | 90.11 | 86.50 | 88.38 | 97.73 | 67.28 | 87.92 | 73.17 |
| | AXIS (N=20%) | 90.32 | 87.05 | 90.48 | 97.59 | 68.91 | 90.65 | 77.37 |
| | AXIS (N=40%) | 89.53 | 86.75 | 92.75 | 96.86 | 70.41 | 92.61 | 78.09 |
| 15 | aTLAS (N=10%) | 91.63 | 86.87 | 78.79 | 98.50 | 66.43 | 85.62 | 68.68 |
| | aTLAS (N=20%) | 92.78 | 87.39 | 82.40 | 98.70 | 67.48 | 87.66 | 70.90 |
| | aTLAS (N=40%) | 92.18 | 87.62 | 84.97 | 98.53 | 67.82 | 89.14 | 72.51 |
| | AXIS (N=10%) | 91.09 | 86.92 | 87.94 | 98.13 | 67.61 | 88.24 | 73.17 |
| | AXIS (N=20%) | 91.03 | 87.64 | 90.54 | 97.89 | 68.70 | 89.97 | 75.87 |
| | AXIS (N=40%) | 90.22 | 87.18 | 92.22 | 97.68 | 70.56 | 92.91 | 77.64 |
| 21 | aTLAS (N=10%) | 92.23 | 87.11 | 80.52 | 98.36 | 66.63 | 86.83 | 70.05 |
| | aTLAS (N=20%) | 92.61 | 87.56 | 81.25 | 98.55 | 66.99 | 87.69 | 71.35 |
| | aTLAS (N=40%) | 92.91 | 88.15 | 84.40 | 98.55 | 67.88 | 89.14 | 73.09 |
| | AXIS (N=10%) | 91.25 | 87.25 | 88.25 | 98.05 | 67.62 | 88.56 | 74.28 |
| | AXIS (N=20%) | 90.81 | 87.46 | 90.86 | 97.95 | 68.96 | 90.38 | 77.35 |
| | AXIS (N=40%) | 90.71 | 86.97 | 91.97 | 97.30 | 70.29 | 92.64 | 79.96 |

Table 12: Detailed results per target dataset for various numbers of source task vectors (STV). Part 3 of 3.

| TV | Strategy | CIFAR100 | CIFAR10 | CUB200 | Caltech101 | Caltech256 | Cars | Country211 | DTD | EuroSAT | FGVCAircraft | Flowers102 | Food101 | GTSRB | MNIST | OxfordIIITPet | PascalVOC | RESISC45 | STL10 | SUN397 | SVHN | UCF101 |
|---|---|---|---|---|---|---|---|---|---|---|---|---|---|---|---|---|---|---|---|---|---|---|
| 1 | bottom components | 76.58 | 95.95 | 58.15 | 94.64 | 88.52 | 64.07 | 17.59 | 67.87 | 97.78 | 29.61 | 79.62 | 85.20 | 87.87 | 97.31 | 91.09 | 86.78 | 88.73 | 98.20 | 66.90 | 84.79 | 72.98 |
|  | top components | 77.00 | 95.85 | 57.61 | 93.89 | 88.44 | 63.54 | 17.70 | 67.02 | 97.30 | 29.70 | 77.49 | 85.81 | 89.57 | 97.36 | 89.92 | 85.77 | 87.51 | 97.65 | 66.80 | 86.63 | 71.00 |
| 2 | bottom components | 77.27 | 95.96 | 58.42 | 94.70 | 88.52 | 63.97 | 17.82 | 65.48 | 97.74 | 29.73 | 79.31 | 85.68 | 88.58 | 97.67 | 91.44 | 87.20 | 88.16 | 98.15 | 67.44 | 86.38 | 74.17 |
|  | arbitrary components | 77.14 | 95.97 | 57.99 | 94.64 | 87.90 | 63.56 | 17.77 | 65.37 | 97.70 | 29.52 | 77.67 | 85.69 | 88.79 | 97.55 | 91.52 | 87.17 | 87.14 | 98.08 | 66.64 | 85.68 | 72.64 |
|  | top components | 77.81 | 96.18 | 57.44 | 93.84 | 87.70 | 63.51 | 17.75 | 67.45 | 96.89 | 29.43 | 75.18 | 85.88 | 90.40 | 97.83 | 89.94 | 86.61 | 88.11 | 97.96 | 66.90 | 87.81 | 73.06 |
| 3 | bottom components | 77.51 | 95.75 | 58.78 | 95.10 | 88.32 | 64.63 | 17.75 | 66.17 | 97.67 | 29.85 | 80.52 | 85.71 | 88.27 | 97.63 | 91.14 | 86.89 | 88.00 | 98.06 | 67.20 | 85.68 | 73.80 |
|  | arbitrary components | 77.46 | 96.12 | 57.90 | 94.41 | 87.85 | 63.79 | 17.88 | 66.17 | 97.93 | 29.49 | 78.06 | 85.54 | 88.36 | 97.57 | 90.76 | 86.36 | 87.73 | 98.10 | 66.43 | 86.51 | 71.85 |
|  | top components | 77.37 | 96.08 | 57.59 | 93.84 | 87.99 | 64.21 | 17.84 | 66.17 | 97.37 | 30.03 | 77.07 | 85.79 | 89.72 | 97.69 | 90.02 | 86.28 | 87.48 | 97.71 | 67.60 | 87.28 | 72.85 |
| 4 | bottom components | 77.36 | 96.11 | 58.35 | 94.47 | 88.37 | 64.18 | 17.91 | 67.02 | 97.63 | 30.30 | 80.19 | 85.77 | 88.61 | 97.75 | 91.11 | 86.87 | 87.37 | 98.21 | 67.11 | 85.56 | 73.25 |
|  | arbitrary components | 77.17 | 96.13 | 58.32 | 94.70 | 87.99 | 63.96 | 17.81 | 66.22 | 97.11 | 30.39 | 79.35 | 85.56 | 88.78 | 97.69 | 91.01 | 86.75 | 87.19 | 98.21 | 67.06 | 86.28 | 73.09 |
|  | top components | 78.32 | 96.01 | 57.89 | 93.15 | 88.32 | 64.23 | 17.91 | 66.86 | 97.85 | 29.82 | 77.72 | 85.88 | 90.32 | 98.00 | 90.11 | 86.79 | 87.41 | 97.84 | 67.35 | 89.26 | 73.80 |
| 5 | bottom components | 77.62 | 95.83 | 57.70 | 94.59 | 88.48 | 64.08 | 17.78 | 65.90 | 97.41 | 29.49 | 78.86 | 85.74 | 88.95 | 97.50 | 90.73 | 86.73 | 88.02 | 98.31 | 66.86 | 84.98 | 72.75 |
|  | arbitrary components | 77.04 | 96.19 | 58.01 | 94.12 | 87.98 | 64.07 | 17.73 | 66.70 | 97.74 | 30.18 | 77.82 | 85.75 | 89.49 | 97.81 | 90.49 | 87.05 | 88.02 | 98.21 | 66.63 | 87.38 | 73.09 |
|  | top components | 77.51 | 96.50 | 58.41 | 93.61 | 88.01 | 63.95 | 18.17 | 65.69 | 97.41 | 30.48 | 77.22 | 86.05 | 90.74 | 97.78 | 90.60 | 86.71 | 87.90 | 97.74 | 67.27 | 90.87 | 71.45 |
| 6 | bottom components | 78.01 | 95.96 | 58.25 | 94.82 | 88.73 | 64.28 | 18.22 | 66.81 | 97.70 | 30.42 | 79.48 | 85.89 | 88.87 | 97.70 | 91.28 | 87.17 | 88.02 | 98.19 | 66.94 | 86.08 | 72.75 |
|  | arbitrary components | 77.54 | 96.17 | 57.99 | 94.53 | 88.08 | 64.21 | 17.82 | 65.32 | 97.89 | 30.45 | 78.48 | 85.77 | 88.43 | 97.50 | 90.84 | 86.69 | 87.92 | 98.05 | 67.09 | 87.58 | 71.42 |
|  | top components | 77.63 | 96.11 | 58.46 | 94.35 | 88.68 | 65.03 | 18.45 | 66.91 | 97.56 | 30.09 | 76.94 | 85.91 | 90.73 | 98.02 | 90.38 | 86.31 | 88.19 | 97.91 | 67.90 | 90.84 | 72.01 |
| 7 | bottom components | 78.04 | 95.93 | 58.44 | 94.87 | 88.52 | 64.51 | 17.77 | 65.37 | 97.59 | 30.07 | 78.37 | 85.94 | 88.57 | 97.39 | 91.28 | 87.07 | 88.02 | 98.08 | 66.88 | 86.09 | 73.43 |
|  | arbitrary components | 77.92 | 95.83 | 57.94 | 95.28 | 87.83 | 64.35 | 17.99 | 65.96 | 97.63 | 28.92 | 78.37 | 85.90 | 88.73 | 97.81 | 90.65 | 86.97 | 87.79 | 98.28 | 66.92 | 85.92 | 73.38 |
|  | top components | 78.12 | 96.20 | 57.99 | 94.41 | 88.22 | 64.72 | 17.82 | 67.61 | 97.81 | 30.51 | 77.54 | 86.15 | 91.43 | 98.36 | 89.97 | 86.53 | 88.22 | 97.80 | 67.43 | 90.80 | 71.95 |
| 8 | bottom components | 78.16 | 95.97 | 58.13 | 94.53 | 88.58 | 64.59 | 17.96 | 65.80 | 97.81 | 30.36 | 80.24 | 85.74 | 88.50 | 97.35 | 91.47 | 86.91 | 87.95 | 98.41 | 67.41 | 86.21 | 73.94 |
|  | arbitrary components | 77.56 | 96.07 | 57.34 | 93.95 | 88.01 | 63.87 | 18.00 | 66.44 | 97.67 | 30.21 | 78.63 | 85.69 | 88.73 | 97.61 | 91.41 | 86.73 | 87.62 | 98.00 | 67.24 | 86.44 | 74.60 |
|  | top components | 79.05 | 96.45 | 58.42 | 93.84 | 88.91 | 64.64 | 18.04 | 66.65 | 97.59 | 30.75 | 79.13 | 86.15 | 91.44 | 98.39 | 90.73 | 86.88 | 88.48 | 97.80 | 67.43 | 90.96 | 73.14 |
| 9 | bottom components | 78.29 | 96.12 | 58.30 | 93.95 | 88.39 | 64.51 | 17.92 | 66.65 | 97.74 | 30.90 | 80.00 | 85.79 | 88.16 | 97.70 | 91.50 | 87.15 | 87.90 | 98.26 | 67.14 | 86.12 | 73.80 |
|  | arbitrary components | 77.95 | 96.13 | 56.94 | 94.64 | 88.50 | 64.11 | 17.85 | 67.45 | 96.96 | 29.43 | 78.50 | 85.86 | 88.49 | 97.85 | 90.57 | 87.03 | 87.84 | 98.28 | 67.11 | 88.73 | 71.58 |
|  | top components | 79.42 | 96.83 | 58.47 | 95.74 | 88.40 | 64.57 | 18.32 | 67.50 | 97.59 | 30.93 | 78.08 | 86.17 | 92.24 | 98.50 | 90.32 | 86.87 | 88.46 | 97.59 | 67.11 | 91.00 | 72.46 |
| 10 | bottom components | 77.86 | 96.12 | 57.49 | 95.10 | 88.66 | 64.36 | 18.03 | 66.01 | 98.00 | 30.37 | 80.37 | 85.95 | 88.28 | 97.59 | 87.13 | 87.56 | 88.11 | 98.11 | 67.39 | 85.85 | 73.49 |
|  | arbitrary components | 77.27 | 96.08 | 57.99 | 94.64 | 88.32 | 63.98 | 17.91 | 65.37 | 97.44 | 30.54 | 79.51 | 85.96 | 88.23 | 97.83 | 90.98 | 87.06 | 87.71 | 98.11 | 67.31 | 85.98 | 72.69 |
|  | top components | 80.09 | 96.96 | 57.85 | 94.82 | 88.76 | 64.66 | 18.24 | 68.14 | 98.00 | 31.95 | 76.65 | 86.15 | 90.02 | 98.02 | 90.11 | 86.50 | 88.38 | 97.73 | 67.28 | 87.92 | 73.17 |
| 11 | bottom components | 77.84 | 95.81 | 57.80 | 93.84 | 88.75 | 64.48 | 18.06 | 67.34 | 97.81 | 30.78 | 80.00 | 85.95 | 87.68 | 97.80 | 91.47 | 87.00 | 88.05 | 98.29 | 67.39 | 86.34 | 73.78 |
|  | arbitrary components | 78.16 | 96.12 | 57.13 | 93.72 | 88.66 | 64.02 | 17.82 | 65.43 | 97.93 | 29.64 | 79.05 | 86.02 | 89.25 | 97.73 | 90.81 | 86.97 | 86.83 | 97.99 | 67.12 | 86.79 | 71.58 |
|  | top components | 80.14 | 96.97 | 59.11 | 95.39 | 88.48 | 64.51 | 18.39 | 66.97 | 97.96 | 30.45 | 78.55 | 86.15 | 90.39 | 98.17 | 90.98 | 87.37 | 87.92 | 97.78 | 67.47 | 88.99 | 74.25 |
| 12 | bottom components | 77.83 | 95.69 | 58.06 | 94.24 | 88.52 | 64.06 | 17.99 | 66.91 | 97.81 | 29.94 | 79.31 | 85.82 | 88.84 | 97.76 | 90.52 | 87.09 | 87.89 | 98.18 | 67.48 | 86.09 | 72.03 |
|  | arbitrary components | 78.18 | 96.29 | 57.58 | 93.89 | 88.44 | 64.22 | 17.75 | 67.66 | 97.85 | 30.93 | 78.18 | 86.01 | 87.66 | 97.72 | 90.92 | 86.89 | 87.43 | 98.08 | 67.06 | 86.60 | 73.62 |
|  | top components | 80.11 | 96.92 | 58.99 | 95.45 | 88.61 | 64.54 | 18.34 | 68.40 | 98.15 | 31.65 | 78.44 | 86.19 | 90.32 | 98.28 | 90.71 | 87.07 | 88.60 | 98.04 | 67.44 | 88.71 | 73.80 |
| 13 | bottom components | 77.82 | 95.89 | 58.53 | 94.18 | 88.86 | 64.12 | 18.20 | 67.55 | 97.93 | 29.70 | 78.73 | 85.90 | 87.64 | 97.57 | 91.50 | 87.07 | 87.84 | 98.31 | 67.42 | 86.25 | 72.72 |
|  | arbitrary components | 78.41 | 95.96 | 58.06 | 95.05 | 88.26 | 64.37 | 18.09 | 66.49 | 97.78 | 29.19 | 79.61 | 85.82 | 88.60 | 97.76 | 90.52 | 86.92 | 87.29 | 98.20 | 67.38 | 86.35 | 72.88 |
|  | top components | 80.05 | 96.83 | 58.85 | 94.70 | 88.81 | 65.10 | 18.38 | 67.39 | 98.19 | 30.84 | 76.81 | 86.17 | 90.28 | 98.18 | 91.36 | 87.26 | 88.54 | 98.05 | 67.55 | 88.84 | 73.62 |
| 14 | bottom components | 77.75 | 95.82 | 58.16 | 94.76 | 88.91 | 64.26 | 18.04 | 67.07 | 97.52 | 29.37 | 78.96 | 85.84 | 88.27 | 97.67 | 90.98 | 86.62 | 87.92 | 98.39 | 67.28 | 85.74 | 72.51 |
|  | arbitrary components | 78.21 | 96.00 | 57.63 | 93.95 | 88.32 | 64.57 | 18.09 | 67.29 | 97.59 | 30.69 | 79.92 | 86.02 | 88.14 | 97.88 | 91.01 | 87.08 | 87.78 | 98.08 | 66.98 | 86.53 | 73.38 |
|  | top components | 80.10 | 96.83 | 58.60 | 95.33 | 88.78 | 64.66 | 18.13 | 67.29 | 98.00 | 30.93 | 77.46 | 86.23 | 90.68 | 98.22 | 91.09 | 87.06 | 88.73 | 98.23 | 67.56 | 88.73 | 73.94 |
| 15 | bottom components | 78.01 | 95.79 | 58.01 | 93.95 | 88.96 | 64.23 | 17.93 | 67.13 | 97.19 | 29.43 | 78.63 | 85.92 | 88.47 | 97.64 | 90.65 | 87.01 | 87.78 | 98.45 | 67.27 | 85.55 | 72.51 |
|  | arbitrary components | 77.94 | 96.24 | 57.27 | 94.53 | 88.71 | 64.28 | 17.79 | 65.90 | 97.85 | 30.33 | 78.66 | 85.97 | 88.90 | 97.72 | 90.73 | 87.02 | 88.24 | 98.21 | 66.95 | 86.53 | 72.98 |
|  | top components | 80.14 | 96.85 | 58.68 | 94.64 | 88.65 | 65.43 | 18.31 | 67.82 | 97.78 | 31.05 | 77.25 | 86.18 | 91.00 | 98.20 | 91.09 | 86.92 | 87.94 | 98.13 | 67.61 | 88.24 | 73.17 |
| 16 | bottom components | 77.94 | 95.90 | 57.73 | 94.30 | 88.48 | 64.66 | 18.21 | 65.32 | 98.00 | 28.98 | 79.74 | 85.93 | 88.27 | 97.56 | 91.06 | 87.13 | 87.65 | 98.28 | 67.39 | 85.97 | 72.32 |
|  | arbitrary components | 78.40 | 96.26 | 57.70 | 94.70 | 88.29 | 64.59 | 18.22 | 67.98 | 97.78 | 30.63 | 80.09 | 86.06 | 88.70 | 97.85 | 91.11 | 86.76 | 88.16 | 97.91 | 67.45 | 87.45 | 72.72 |
|  | top components | 80.13 | 96.78 | 59.06 | 94.47 | 88.88 | 65.30 | 18.28 | 68.09 | 97.74 | 30.63 | 78.37 | 86.23 | 90.58 | 98.03 | 90.79 | 87.16 | 87.76 | 97.91 | 67.62 | 88.64 | 74.31 |
| 17 | bottom components | 77.64 | 95.87 | 57.85 | 94.99 | 88.30 | 64.40 | 17.89 | 66.49 | 97.52 | 30.45 | 80.47 | 85.99 | 88.18 | 97.78 | 90.73 | 87.35 | 88.10 | 98.35 | 66.93 | 85.91 | 72.64 |
|  | arbitrary components | 78.15 | 96.27 | 57.90 | 94.47 | 88.14 | 64.61 | 17.97 | 65.74 | 97.59 | 30.06 | 79.35 | 86.03 | 88.47 | 97.70 | 91.25 | 87.23 | 87.78 | 98.21 | 66.98 | 86.42 | 73.20 |
|  | top components | 80.17 | 96.91 | 58.77 | 95.45 | 88.89 | 65.03 | 18.27 | 66.76 | 97.89 | 30.12 | 78.96 | 86.23 | 90.82 | 98.10 | 91.20 | 86.75 | 87.75 | 97.95 | 67.67 | 88.38 | 74.31 |
| 18 | bottom components | 77.55 | 96.01 | 58.47 | 94.18 | 88.91 | 64.71 | 17.88 | 66.81 | 98.04 | 29.37 | 80.89 | 85.93 | 89.04 | 97.51 | 90.65 | 87.09 | 87.68 | 98.39 | 67.50 | 86.62 | 72.22 |
|  | arbitrary components | 78.34 | 96.14 | 57.85 | 94.07 | 88.37 | 65.10 | 18.05 | 66.76 | 97.44 | 29.04 | 80.68 | 86.10 | 88.92 | 97.77 | 91.41 | 87.23 | 87.27 | 98.36 | 67.20 | 86.65 | 73.22 |
|  | top components | 80.00 | 96.86 | 58.70 | 94.87 | 88.91 | 65.10 | 18.22 | 68.78 | 97.74 | 31.95 | 77.74 | 86.17 | 90.67 | 98.37 | 91.20 | 86.79 | 87.89 | 97.89 | 67.58 | 88.29 | 75.05 |
| 19 | bottom components | 77.76 | 95.92 | 58.56 | 94.18 | 88.47 | 64.43 | 17.91 | 66.44 | 97.89 | 30.57 | 79.85 | 85.85 | 88.85 | 97.68 | 90.43 | 87.25 | 88.03 | 98.44 | 67.45 | 85.51 | 72.32 |
|  | arbitrary components | 77.71 | 95.87 | 58.32 | 94.82 | 88.58 | 65.00 | 17.95 | 66.28 | 97.89 | 29.79 | 79.28 | 85.86 | 89.69 | 98.07 | 90.68 | 86.99 | 87.97 | 98.20 | 67.05 | 86.93 | 72.91 |
|  | top components | 79.98 | 96.89 | 59.15 | 95.22 | 88.97 | 64.86 | 18.46 | 67.87 | 98.07 | 32.40 | 77.82 | 86.25 | 90.89 | 98.07 | 90.98 | 87.19 | 88.33 | 98.05 | 67.79 | 88.18 | 74.73 |
| 20 | bottom components | 77.62 | 96.04 | 58.61 | 94.12 | 88.45 | 64.81 | 17.94 | 66.54 | 98.00 | 29.91 | 79.67 | 85.88 | 88.65 | 97.84 | 91.09 | 87.25 | 88.02 | 98.44 | 66.89 | 86.67 | 72.96 |
|  | arbitrary components | 77.72 | 95.99 | 57.80 | 94.47 | 88.29 | 65.08 | 17.86 | 66.91 | 97.56 | 29.88 | 79.46 | 86.23 | 88.73 | 97.88 | 91.09 | 86.79 | 87.38 | 98.24 | 66.89 | 86.42 | 72.48 |
|  | top components | 80.07 | 96.93 | 58.94 | 95.10 | 88.88 | 64.83 | 18.44 | 68.94 | 97.70 | 30.00 | 78.03 | 86.35 | 90.69 | 98.11 | 91.14 | 87.37 | 88.29 | 97.99 | 67.65 | 88.30 | 74.86 |
| 21 | bottom components | 77.57 | 95.85 | 58.63 | 94.07 | 88.65 | 64.74 | 18.19 | 65.37 | 97.48 | 30.30 | 79.74 | 85.97 | 87.06 | 97.82 | 90.84 | 87.23 | 88.46 | 98.46 | 67.52 | 85.48 | 73.28 |
|  | arbitrary components | 78.05 | 96.16 | 57.61 | 94.99 | 87.98 | 64.82 | 17.98 | 66.49 | 97.22 | 29.64 | 77.65 | 86.08 | 88.52 | 97.55 | 90.60 | 86.96 | 87.38 | 98.28 | 67.11 | 87.73 | 73.88 |
|  | top components | 80.11 | 96.93 | 58.46 | 94.99 | 88.76 | 65.09 | 18.48 | 67.98 | 97.81 | 30.75 | 77.87 | 86.32 | 91.06 | 98.11 | 91.25 | 87.25 | 88.25 | 98.05 | 67.62 | 88.56 | 74.28 |

Table 13: A detailed, per-dataset performance comparison of different SVD component aggregation strategies. The table reports the Top-1 accuracy (%) for each target task, illustrating how performance evolves as the number of aggregated source task vectors (TV) increases. We compare our primary top components strategy against bottom components and arbitrary components as baselines to validate the robustness of our selection method across diverse data domains.

