# OpenReview forum: "Efficient Multi-Source Knowledge Transfer by Model Merging"
_ICLR.cc/2026/Conference — Submitted to ICLR 2026_

### Official Review · Reviewer_LXNa · 2025-10-26

**Soundness:** 2
**Presentation:** 2
**Contribution:** 2
**Rating:** 4
**Confidence:** 3

**Summary:**

This paper proposes AXIS, a model-merging framework designed to efficiently integrate knowledge from multiple fine-tuned models for transfer to new target tasks. AXIS leverages Singular Value Decomposition (SVD) to decompose and aggregate task-specific models trained on different datasets, enabling more effective adaptation to unseen tasks. By fine-tuning only the singular values of the merged model, AXIS efficiently consolidates essential knowledge from multiple source tasks while significantly reducing storage and computational costs.

**Strengths:**

1. The SVD-based rank-1 component selection enables fine-grained and interpretable model merging, substantially improving over traditional fusion methods.

2. The one-time aggregation stage effectively consolidates knowledge, while subsequent fine-tuning operates on a fixed-size matrix, ensuring constant training and memory costs regardless of the number of source models.

3. The method demonstrates robustness under challenging conditions.

4. Comprehensive experiments and comparisons with aTLAS validate the effectiveness and efficiency of the proposed AXIS framework.

**Weaknesses:**

1. The proposed method assumes that all source models share the same architecture and originate from the same pre-trained backbone, which is a restrictive and unrealistic assumption. The approach cannot currently merge models of different architectures (e.g., CNNs) or even different scales of Vision Transformers. This limits its applicability in heterogeneous or cross-architecture settings—arguably the most promising real-world use cases.

2. Although the paper provides extensive experiments, it only compares AXIS with aTLAS. This comparison set is not sufficient to fully establish the method’s advantages. Including other representative baselines—such as knowledge distillation, model stitching, or parameter-efficient fine-tuning (PEFT) approaches—would make the evaluation more convincing.

3. All experiments are conducted on relatively small-scale vision datasets, which share high inter-task similarity. Evaluating on larger-scale datasets or across modalities (e.g., text or multimodal benchmarks) would better test the generalization and scalability of AXIS. Especially in NLP tasks, where task heterogeneity is more pronounced, the benefits of multi-source knowledge integration might be more clearly demonstrated.

4. While SVD provides a solid mathematical foundation, the choice of Top-K singular components is based primarily on empirical intuition. The paper would be strengthened by theoretical or empirical analysis demonstrating why high-magnitude components consistently correspond to transferable knowledge.

5. The overall layout of this article is a bit messy.

**Questions:**

1. Could low-magnitude singular components still contain task-specific or complementary knowledge that may be useful for transfer?

2. Do the selected singular vectors possess interpretable semantics across tasks? Some visualization or qualitative analysis on vision datasets would help illustrate this.

3. How could AXIS be extended to support cross-scale or cross-architecture model merging?

4. Can the method be applied or adapted to textual or multimodal datasets to demonstrate broader generality?

5. Although AXIS achieves parameter efficiency, the FLOPs are not significantly reduced. Could the authors explore further compression to achieve both effective knowledge merging and efficient deployment?

---

> ### Author Response · Authors · 2025-11-28
> **Official Comment by Authors**
>
> Thank you for your detailed and constructive review.
>
> **Are low-magnitude components relevant to transferability?**
>
> To answer this question, we conducted two experiments. In the first, we fine-tuned one singular value from $\Delta_m$ at a time, starting with the value of highest magnitude. The results, averaged over all target tasks, reveal a downward trend: the top-ranked component yields the highest performance, at 61.64% accuracy, followed by a progressive decline for lower-ranked components, dropping to approximately 30–40% (see Fig. 11 in the Appendix).
> The second experiment measures the similarity between the matrices generated by each component of $\Delta_m$  and the matrix obtained by fine-tuning the model for the target task (upper bound, not available during training). As shown in Fig. 12 of the Appendix, the matrices of the top-ranked components are more similar to the target matrix, which justifies using magnitude as a proxy for transferability. See the “Transferability by SVD components” section of the Appendix for details.
>
> **Could AXIS be extended to support merging models across scales or architectures?**
>
> Cross-scale transfer (ViT-B-32 to ViT-B-16): In our new experiments, we transferred knowledge between models with different patch sizes by mapping compatible layers and skipping those that were dimensionally incompatible (only two layers). The result (79.16% accuracy) was very close to that of the same-architecture baseline (80.59%), proving that the AXIS is robust to architectural variances. Additional results are provided in the "Investigating transfer boundaries" section of the Appendix. Merging fundamentally distinct architectures (e.g. combining CNNs with ViTs) is beyond the scope of our work.
>
> **Is AXIS applicable to natural language processing (NLP) or multimodal tasks?**
>
> We ran additional experiments on language models with T5-Base architecture across seven datasets: PAWS, QASC, Quartz, Story Cloze, WikiQA, Winogrande, and WSC. We conduct an exhaustive evaluation that demonstrates the strong effectiveness of AXIS in the NLP domain (see Fig.10 of the Appendix), including that AXIS outperforms aTLAS across all settings. We present findings within the new 'Performance on T5-Base language models' section in Appendix. Finally, we believe AXIS facilitates bidirectional knowledge transfer between vision and text tasks, though we leave this exploration for future work.
>
> **Can AXIS be compressed any further for efficient deployment?**
>
> Since AXIS produces a weight matrix that complies with the base architecture, the number of floating-point operations (FLOPs) required for inference remains unchanged compared to the base model. However, this design ensures full compatibility with standard post-training compression techniques (e.g. quantisation or pruning) should further compression be required for deployment.
>
> **Is the evaluation sufficient without baselines such as distillation or stitching?**
>
> Our evaluation encompasses a broader set of baselines than just aTLAS, incorporating both PEFT methods and alternative model merging techniques:
> 1. Comparison with PEFT: as illustrated in Fig. 5, we have conducted a direct comparison between AXIS and the widely adopted PEFT approaches LoRA and LoRA-XS. The results demonstrate that AXIS achieves a superior performance-to-parameter trade-off compared to these methods (for example, AXIS consistently outperforms LoRA across all parameter budgets), thus validating the effectiveness of reusing merged weights over starting adaptation from pre-trained weights alone.
> 2. Comparison with other merging methods (Fig. 4 and Table 1): Beyond aTLAS, we evaluated our adaptation stage on other state-of-the-art merging baselines, including DARE, TIES-Merging and TSV-M.
> 3. Although we recognise the value of Knowledge Distillation and Model Stitching, they refer to different problems of machine learning, and are not considered for comparison in other papers in the field of model merging.

---

### Official Review · Reviewer_R9qd · 2025-10-27

**Soundness:** 2
**Presentation:** 3
**Contribution:** 2
**Rating:** 4
**Confidence:** 2

**Summary:**

This paper proposes a method for parameter-efficient fine-tuning. It integrates the knowledge from multiple source models which are fine-tuned on different downstream tasks. These fine-tuned models (represented by their parameter difference from the pre-trained model) serve as the knowledge source for the target task. SVD is performed for each parameter difference, and the top-K components with highest singular values are selected to reconstruct the merged difference matrix $\Delta_m$, which is further used as the starting point of target task fine-tuning. The second stage involves fine-tuning on top singular values after decomposing the sum of the pre-trained parameter and $\Delta_m$. Experiments on a diverse set of benchmarks are conducted to compare the proposed method AXIS and aTLAS. Results show that AXIS not only achieves overall higher accuracy in terms of test accuracy, but also requires less computation and memory cost, especially when the number of task vectors increases.

**Strengths:**

1. The proposed idea is interesting and achieves better empirical results compared to the baseline aTLAS on both accuracy and complexity.
2. The paper is well written. The method is described clearly.
3. The analytical experiments help readers to understand how the important factors affect the performance of the algorithm.

**Weaknesses:**

1. Despite the good performance, the motivation and rationale of the proposed method seem not fully convincing. Particularly, why is the most transferable useful knowledge for the target task represented by the principal singular components (as stated in line 160-161)? Is there any theoretical insight or empirical evidence for this? I didn’t find explanations for this, but the analytical results seem to be conflict with the motivation stated in line 160-161. Specifically, results in Table 1 show that a simple averaging for Stage 1 already shows good enough results (especially for N=10%/20%, and all results are much better than aTLAS), implying that the singular value fine-tuning in Stage 2 is the dominant factor for the success of the proposed method rather than the low-rank reconstruction. Table 2 also shows that Arbitrary and Bottom components are almost as good as the top components.
2. While the Stage 2 strategy seems to contribute a lot to the whole framework, the idea of singular value fine-tuning has already been explored [1].

[1] Singular Value Fine-tuning: Few-shot Segmentation requires Few-parameters Fine-tuning. (NeurIPS 2022)

**Questions:**

1. Why does AXIS achieve higher accuracy than aTLAS with less runtime and memory?
2. How did you get the source task vectors described in line 238? Are they from the same datasets used in the experiments or external datasets?
3. In Figure 11, why does the performance of the w/o group drop dramatically when using a moderate number of task vectors?

---

> ### Author Response · Authors · 2025-11-28
> **Official Comment by Authors**
>
> Thank you for your detailed and constructive review.
>
> **Does fine-tuning the principal singular components bring the most transferable knowledge?**
>
> We conducted two additional experiments to answer this question. In the first one, we fine-tuned only one singular value from $\Delta_m$ at a time, starting with the one of highest magnitude. Results averaged over all target tasks reveal a downward trend: the top-ranked component yields the highest performance, at 61.64% accuracy. This is followed by a progressive decline for lower-ranked components, dropping to approximately 30–40% (see Fig. 11 in the Appendix).
> The second experiment measured the similarity between the matrices generated by each component of  $\Delta_m$ and the matrix obtained by fine-tuning the model for the target task (upper bound, not available during training). As shown in Fig. 12 of the Appendix, the matrices of the top-ranked components are more similar to the target matrix, which justifies using magnitude as a proxy for transferability. We have included a new section in the appendix “Transferability by SVD components”, in which we provide direct empirical evidence and analysis to validate this design choice.
>
> **Why Stage 1 cannot be replaced with another merging method?**
>
> We demonstrate that Stage 1 is essential for creating a robust, high-quality transfer basis in two challenging scenarios that test AXIS robustness: (1) When we randomly mask the input with different dropout rates, AXIS outperforms other merging baselines and AXIS variants, as shown on the left side of the updated Fig. 4; (2) With the corrupted source the standard merging methods are vulnerable to low-quality sources as they incorporate corrupted parameters blindly. In contrast, as presented on the right side of Fig. 4, AXIS is highly resilient to this corruption. Thus, even when followed by Stage 2 adaptation, other methods suffer significant performance degradation in these realistic scenarios. Hence, the AXIS combination of Stage 1 and Stage 2 yields the best overall results.
>
> **Why does AXIS achieve higher accuracy than aTLAS in less time and with less memory?**
>
> aTLAS requires the entire set of T source models to be loaded and accessible during the forward and backward passes in order to learn the mixing coefficients. Consequently, its memory and runtime costs scale linearly (O(T)). In AXIS, knowledge aggregation is performed as a one-time step, which merges all T sources into a single, compact merged matrix ($\Delta_m$). The training phase (Stage 2) then operates on this merged matrix, regardless of the number of source models. Hence, the adaptation cost for AXIS is constant (O(1)).
>
> Besides these pure algorithmic differences, aTLAS computes a linear combination of entire task vectors, including task-specific noise and redundancy alongside features. In contrast, AXIS uses SVD to decompose parameters into elementary rank-1 components. By selecting the Top-K principal components, the less useful 'tail' of the spectrum is explicitly filtered out. Moreover, in Stage 2, AXIS freezes the singular vectors and only fine-tunes the singular values. This preserves the robust structural priors learned during the aggregation phase. In contrast, aTLAS’s full-parameter combinations are more prone to overfitting and distorting these beneficial directions during adaptation.
>
> **How were the source task vectors obtained?**
>
> The source task vectors do not come from external datasets. We used the public library of task vectors provided by the aTLAS authors, which were obtained by fine-tuning the pre-trained CLIP model on distinct image recognition datasets. We describe them in the “Experimental setup” section of the original Appendix.
>
> **Why does the performance of the w/o group drop dramatically when using a moderate number of task vectors (Fig. 11 of the original paper)?**
>
> We do not have strong evidence why it is happening. However, we have some intuition on that, based on the fact that when we use singular vectors directly from $\Delta_i$, some of them can be linearly dependent. Consequently, updating the singular value of one component collides with other linearly dependent components. For a moderate number of tasks, the ratio of vector collisions is higher than for larger numbers of tasks. That can be the reason for the performance drop in the former.
>
> **Difference between AXIS and paper [1]**
>
> Paper [1] uses SVD for few-shot segmentation tasks. However, it differs from AXIS for the following reasons:
> 1. Paper [1] applies SVD directly to the pre-trained weights ($\theta_{pre}$). AXIS differs fundamentally in that it uses SVD for (a) aggregating important components from many source models and (b) adapting the merged task matrix ($\Delta_m$) to the target task.
> 2. AXIS achieves superior efficiency, as the method in [1] corresponds to the 'Pre-trained TV' baseline (see Fig. 5), which performs worse than AXIS.

---

### Official Review · Reviewer_eMxV · 2025-10-31

**Soundness:** 2
**Presentation:** 1
**Contribution:** 2
**Rating:** 4
**Confidence:** 4

**Summary:**

This paper proposes AXIS, a two-stage framework for multi-source transfer via SVD-based component extraction and aggregation, followed by parameter-efficient adaptation. AXIS reports higher average accuracy and improved variance, constant memory/runtime during adaptation, and robustness to noisy/pruned task vectors and masked-patch inputs.

**Strengths:**

1. Clear component-level view of model merging: extract rank-1 atoms across many sources, globally rank, then merge and adapt, which is a granular alternative to full-vector merging such as aTLAS.

2. If broadly valid, AXIS provides a recipe for fusing many fine-tuned models into a compact, adaptable basis, which is useful for multi-source transfer with predictable compute.

**Weaknesses:**

1. In lines 98-101, the authors claimed that 'While numerous works explore combining models’ weights with diverse architectures..., these often prove less effective than approaches that assume all considered models originate from the same base model'. I would say that these two lines of work target different scenarios. Using the shortcomings of one family of methods as the reason for choosing the other in its own application setting is odd and does not constitute a sound motivation.

2. The definition is confusing. In line 131, task vector is defined as τi, and in line 143, ∆i is used to define task matrix. But in line 152, ∆i is used to referred as task vectors.

3. In Figure 10, the results indicate that AXIS scales consistently with the number of trained parameters. In the case N% = 100%, how does the method perform? Based on the trend, performance appears to surpass standard full-parameter fine-tuning (the baseline), which seems counterintuitive. Where exactly does this performance gain come from?

4. When confronted with multiple unseen tasks, the method appears to require restarting training for each task. Can the authors propose any improvements to training efficiency in this setting? Also, What is the transferability of the learned matrix across tasks? how well does the learned AXIS perform when directly applied to other unseen tasks without retraining?

5. The choice of K in K% appears to rely heavily on post-hoc judgment and manual selection.

6. The manuscript contains substantial unused white space (e.g., the right side of column 96, the area above Fig. 7, the space below Table 1, and below Fig. 10). Please tighten the layout and ensure figures/tables are placed without large gaps. Also, there are several typos, such as line 219 (‘algorithm ?? and figure 2’).

7. Please give a compact, layer-wise definition (matrices vs. biases) and a consistent symbol table, and clarify how non-matrix params are handled (you mention averaging). A small per-layer example would help.

**Questions:**

See weakness. I'm willing to increase the rating once my concerns are well addressed.

---

> ### Author Response · Authors · 2025-11-28
> **Official Comment by Authors**
>
> Thank you for your detailed and constructive review. We corrected the mentioned errors in the article and improved the presentation quality.
>
> **Clarification on motivation**
>
> As noted by the reviewer, merging heterogeneous and homogenous architectures are distinct research scenarios. However, we did not intend to weaken the importance of cross-architecture merging, but rather to emphasize that we focus on a widely applicable setting of shared initialization. We concentrated on this scenario due to the vast availability of diverse fine-tuned models derived from popular foundation models. We rephrased the Related Works section to match our intention.
>
> **Is AXIS robust enough to bridge the gap in the cross-initialization setting?**
>
> While our primary focus is on the shared-initialization setting, we ran additional experiments, and found that AXIS is robust outside these strict constraints. As described in new Appendix section "Investigating transfer boundaries", our method performs efficiently even when transferring task vectors between different initializations (e.g., ViT-B-32 to ViT-B-16) or distinct pre-training datasets (OpenCLIP) - see Table 6 of the Appendix.
>
> **Does AXIS surpass the standard full fine-tuning baseline?**
>
> The results shown in Table 7 of the Appendix show that AXIS with accuracy of 82.30% (for N=100% corresponds to fine-tuning all singular values) closely approaches but does not surpass the standard full fine-tuning with 83.56% accuracy. However the latter requires dataset-specific hyperparameters and longer training (ranging up to 76 epochs), whereas AXIS uses a fixed budget of only 10 epochs for all datasets and a negligible fraction of the total model parameters.
>
> **How much the choice of K in Top-K relies on manual tuning rather than a systematic selection method?**
>
> As shown in the sensitivity analysis in Fig. 6 of the original paper, AXIS maintains high performance across a wide range of K values. The accuracy drop is minimal (less than 1.5%) between K values, indicating stability with respect to this hyperparameter. Additionally, we conducted an experiment in which Optimal Hard Thresholding (Gavish & Donoho, 2014) was used to automatically determine K for each matrix based on noise estimation. However, this theoretically grounded method did not improve performance over our approach. That is why, we decided to use fixed K, similarly to other well-established multi-task merging methods [2, 3].
>
> **Can the method efficiently handle multiple new tasks without requiring retraining for each of them?**
>
> To answer this question, we ran an additional experiment that tests AXIS with a joint multi-task adaptation strategy. In this strategy, the AXIS model had six task-specific heads and was trained simultaneously with a mixture of data from all those tasks. As shown in the Fig. 13 in the 'Multi-task performance of AXIS' section of the Appendix, joint training results in a performance drop of approximately 3-4% compared to the specialised, individual AXIS baselines (e.g. 67% vs. 70.5%). Hence, although AXIS supports joint training, it would require the integration of multi-task optimisation techniques to achieve an optimal performance.
>
> **Does the fine-tuned AXIS model demonstrate zero-shot transferability to unseen tasks?**
>
> To answer this question, we conducted experiments in which we trained AXIS individually to six different targets, and tested it on five remaining ones in a zero-shot manner. As presented in Fig. 14 of the Appendix, AXIS do not demonstrate zero-shot transferability. This is expected behaviour because its primary design objective is high-fidelity specialisation to the target domain.
>
> [1] Eckart, C., & Young, G. (1936). The approximation of one matrix by another of lower rank. Psychometrika, 1(3), 211-218
>
> [2] Gargiulo, A. A., Crisostomi, D., Bucarelli, M. S., Scardapane, S., Silvestri, F., & Rodola, E. (2025). Task singular vectors: Reducing task interference in model merging. In Proceedings of the Computer Vision and Pattern Recognition Conference (pp. 18695-18705).
>
> [3] Marczak, D., Magistri, S., Cygert, S., Twardowski, B., Bagdanov, A. D., & van de Weijer, J. (2025). No task left behind: Isotropic model merging with common and task-specific subspaces. arXiv preprint arXiv:2502.04959.

---

> ### Author Response · Authors · 2025-11-28
> **Official Comment by Authors**
>
> **Clarification on $\tau_i$ and $\Delta_i$ definitions**
>
> Section 3.1 has been revised to explicitly distinguish between general task vectors (denoted by  $\tau_i$) and 2D task matrices suitable for SVD (denoted by $\Delta_i$).
>
> **Clarification on notation and parameter-specific aggregation**
>
> We use the following extended notation in Appendix:
> | Symbol | Dimensionality | Definition |
> | :--- | :--- | :--- |
> | $\theta_{pre}^{(l)}$ | Any shape | Parameters of the pre-trained base model at layer $l$. |
> | $\theta_{i}^{(l)}$ | Any shape | Parameters of the fine-tuned model for source task $i$ at layer $l$. |
> | $\tau_{i}^{(l)}$ | Any shape | Task vector defined as $\tau_{i}^{(l)} = \theta_{i}^{(l)} - \theta_{pre}^{(l)}$. |
> | $\Delta_{i}^{(l)}$ | $m \times n$ | Task matrix: A specific subset of task vector $i$ corresponding to one of 2D matrices from layer $l$. |
>
> Our framework treats parameters differently based on their structural dimensionality:
> - For matrices, the task difference is treated as a matrix $\Delta_{i}^{(l)}$ to which we apply the AXIS algorithm.
> - For biases, we average parameters across all source tasks.

---

### Official Review · Reviewer_uXKj · 2025-11-04

**Soundness:** 3
**Presentation:** 3
**Contribution:** 3
**Rating:** 6
**Confidence:** 4

**Summary:**

This paper proposes the AXIS framework to address limitations in multi-source knowledge transfer, such as the coarse granularity, low efficiency, and poor robustness of existing methods like aTLAS.
AXIS follows a two-stage workflow. In the first stage (knowledge extraction and aggregation), each source task matrix is decomposed using Singular Value Decomposition to obtain elementary components. It then selects the most salient top-K components through global ranking and aggregates them into a single merged matrix. In the second stage (target task adaptation), the merged matrix undergoes Singular Value Decomposition again. Only the top-N principal singular values are fine-tuned for the target task, while other components remain frozen to ensure parameter efficiency.
Experiments on 21 image classification tasks using ViT-B-32 and ViT-L-14 architectures show that AXIS outperforms aTLAS in accuracy while maintaining constant memory usage and runtime. Moreover, AXIS demonstrates greater robustness against source parameter noise or sparsity and is more resilient to input degradation. It also surpasses parameter-efficient fine-tuning (PEFT) methods such as LoRA—all while relying on the same shared pre-trained model architecture.

**Strengths:**

AXIS resolves the efficiency bottleneck of aTLAS where memory and runtime increase linearly with the number of source tasks by splitting multi-source knowledge transfer into two stages: knowledge extraction and aggregation, and target task adaptation. This ensures that memory and runtime costs do not grow with the increase in the number of source models.
It can filter out unstructured noise and redundancy by screening the Top-K significant components of the multi-source task matrix through singular value decomposition, and is superior to aTLAS in robustness.

**Weaknesses:**

AXIS fundamentally assumes that all source models share the same architecture as the pre-trained model and are fine-tuned from a common pre-trained base; thus, it is inapplicable when source models are architecturally heterogeneous (e.g., a mix of CNNs and ViTs) or derived from different pre-trained models, severely limiting its use in cross-architecture or cross-pretraining multi-source knowledge transfer scenarios.
It performs knowledge aggregation as a one-time offline operation; thus, adding new source models requires recomputing SVD decomposition, top-K selection, and aggregation over all sources (both old and new), making it non-incremental and unsuitable for real-time scenarios with dynamically expanding source models.

**Questions:**

The experiment only focused on the image classification task and was based on the ViT architecture. It did not extend to other computer vision tasks such as object detection and semantic segmentation, nor did it verify the effectiveness in cross-modal scenarios such as NLP and speech. I want to know whether it has generalization ability in non-image classification and non-VIT architecture

---

> ### Author Response · Authors · 2025-11-28
> **Official Comment by Authors**
>
> Thank you for your detailed and constructive review.
>
> **Is AXIS limited to Vision Transformers and image classification?**
>
> We ran additional experiments on language models with T5-Base architecture across seven datasets: PAWS, QASC, Quartz, Story Cloze, WikiQA, Winogrande, and WSC. We conduct an exhaustive evaluation that demonstrates the strong effectiveness of AXIS in the NLP domain (see Fig.10 of the Appendix), including that AXIS outperforms aTLAS across all settings. We present findings within the new 'Performance on T5-Base language models' section of the Appendix.
> Regarding non-ViT architectures, AXIS can merge CNNs by performing SVD directly on the convolutional filters.
>
> **In paper, AXIS assumes that all fine-tuned models have a common pretrained base with fixed architecture. Does AXIS work in cross-architecture or cross-pretraining scenarios?**
>
> Merging fundamentally distinct architectures (e.g., combining CNNs with ViTs) is outside the scope of our work. However, AXIS can be applied to different pretrained bases or variants of architecture. We conducted a new suite of experiments, where base model ViT-B-16 were merged with task vectors obtained from: (1) the same model fine-tuned on Cars; (2) ViT-B-32 fine-tuned on Cars; (3) ViT-B-16 pretrained on OpenCLIP and fine-tuned on Cars; (4) BiomedCLIP (ViT-B-16 pretrained on OpenCLIP and fine-tuned on biological data).
> Merging with a task vector obtained for a different architecture variant (option 2) provides a small performance drop (from 80.59% to 79.41%). Similarly, like merging with a task vector obtained for a different pretraining dataset (option 3), for which the accuracy dropped to 79.16%. In contrast, in the case of BiomedCLIP, which is out of distribution model (option 4), the drop is much larger (13.62%). Detailed results are provided in section "Investigating Transfer Boundaries" of the Appendix.
>
> **Does adding a new source model require a full re-execution of the entire pipeline (SVD, ranking, aggregation)?**
>
> While global component ranking from all sources is the default option in the paper, the merged matrix can be updated incrementally by processing incoming sources sequentially. To demonstrate this, we evaluate AXIS using a dynamic, incremental aggregation protocol during Stage 1 from Fig. 2 of the paper. Starting with an initial merge of two vectors, we iteratively fuse each new incoming source task vector directly into the single, evolving consolidated model. Source parameters are discarded immediately after integration, and no buffer of historical models is maintained. This design evaluates efficient online knowledge accumulation in Stage 1, after which the final merged model is adapted to the target domain. Results averaged over all target tasks within a single run indicate that the accuracy of AXIS with incremental aggregation is 78.55%, which is similar to the accuracy of 78.48% obtained for the standard AXIS protocol presented in the paper. We added this ablation study to the Appendix section 'Incremental knowledge aggregation'.

---

### Official Review · Reviewer_Xmc5 · 2025-11-11

**Soundness:** 2
**Presentation:** 2
**Contribution:** 3
**Rating:** 4
**Confidence:** 3

**Summary:**

The paper introduces AXIS, a two-stage merge-and-tune paradigm for multi-source transfer learning. In the first stage, each source task matrix is decomposed via singular value decomposition (SVD), and rank-one components from all sources are aggregated, selecting the top-K based on their singular values to form a compact merged task matrix. In the second stage, only the singular values of this merged matrix are fine-tuned on the target task, while the singular vectors remain fixed. The authors claim that this design achieves greater performance, efficiency, scalability, and robustness compared to the existing aTLAS framework. Extensive experiments across a wide range of classification benchmarks support these claims.

**Strengths:**

1. The paper points out a highly relevant problem in modern deep learning, i.e. how to leverage the immense and growing number of specialized models publicly available. The failure of standard transfer learning to utilize this "wealth of specialized knowledge"  is a clear and well-motivated gap.
2. Strong empirical results on scalability (Fig. 5) and robustness (Figs. 6-7) support the paper’s key claims of efficiency and resilience.
3. Comprehensive ablation studies are conducted, examining sensitivity to K (number of components), N (trainable singular values), selection strategies (top / arbitrary / bottom), and the impact of the final re-SVD orthogonalization. These analyses effectively illustrate why the method works and where its performance gains originate.

**Weaknesses:**

1. The manuscript requires careful proofreading to improve overall presentation quality. There are numerous minor issues that detract from readability. For example, Figure 3 is not referenced in the text, "Algorithm ??" appears in line 219 due to a missing reference, and the Related Works section begins with an incomplete line. Addressing these formatting and referencing errors would significantly improve clarity and professionalism.
2. The use of a radar plot in Figure 3 makes it difficult to clearly assess the claimed state-of-the-art performance. While it provides a compact visualization, it hides exact numerical differences across datasets and even suggests that AXIS underperforms the previous SOTA on several tasks. A detailed table of per-dataset results (e.g., accuracy or relative gain) would provide much clearer evidence and allow fairer comparison with baselines.
3. The decision to fine-tune only the singular values in the second stage is parameter-efficient, but fixing the singular vectors may limit adaptation flexibility, as vectors from diverse sources might not align well with the target task directions. While the authors partially justify this design through Stage 2 effectiveness experiments, a stronger empirical comparison would provide a more complete validation.
4. The authors explicitly state AXIS relies on a common architecture and shared pre-trained initialization in conclusion. This restriction weakens applicability in realistic model zoos where many models differ by initialization, minor architectural variants, or LoRA-style adapters.
5. Limited justification for selecting by raw singular value magnitude as a proxy for transferability is provided. While experiments show it works empirically, the paper lacks theoretical or empirical analysis connecting singular value size to task transferability.

**Questions:**

Same as weaknesses

---

> ### Author Response · Authors · 2025-11-28
> **Official Comment by Authors**
>
> Thank you for your detailed and constructive review. We have corrected the mentioned errors in the article and improved the presentation quality.
>
> **Why do we fine-tune only singular values instead of singular vectors?**
>
> The reason for fine-tuning singular values ($\Sigma_t$) instead of singular vectors ($U_t, V_t$)  lies in the vast majority of parameters in the latter. For example, in a standard ViT-B-16 projection layer (with $768 \times 3072$ parameters), the singular vectors account for nearly 2.95 million parameters, while there are less than 800 parameters of singular values.
> To provide empirical evidence on the effectiveness of fine-tuning singular values, we ran an experiment, where we randomly chose the same number of parameters N from singular vectors and fine-tuned them instead. As a result, we obtain a drastic decrease in accuracy (from 84.11% to 57.41%).
> We additionally analyzed what will happen if we fine-tune random parameters of the matrix $\Sigma_t$ instead of only its diagonal. The decrease of accuracy is smaller, but still significant (from 84.11% to 81.55%).
> In summary, the experiments show that it is more effective to fine-tune singular values instead of singular vectors. We present detailed results in Table 5 of the Appendix.
>
> **In paper, AXIS relies on a common architecture and shared pre-trained initialization. But, can it merge models in more demanding scenarios, where models have different initialization, architecture variants, or LoRA adapters?**
>
> To answer this question, we conducted a new suite of experiments, where base model ViT-B-16 were merged with task vectors obtained from: (1) the same model fine-tuned on Cars; (2) ViT-B-32 fine-tuned on Cars; (3) ViT-B-16 pretrained on OpenCLIP and fine-tuned on Cars; (4) BiomedCLIP (ViT-B-16 pretrained on OpenCLIP and fine-tuned on biological data).
> Merging with a task vector obtained for a different architecture variant (option 2) provides a small performance drop (from 80.59% to 79.41%). Similarly, like merging with a task vector obtained for a different pretraining dataset (option 3), for which the accuracy dropped to 79.16%. In contrast, in the case of BiomedCLIP, which is out of distribution model (option 4), the drop is much larger (13.62%). We conclude that AXIS could be used in realistic model zoos.
> Considering the question about LoRA, we would like to note that once LoRA is reconstructed into one matrix $\Delta W = BA$, it becomes mathematically equivalent to $\Delta_m$ from Algorithm 1 of the paper.
> Detailed results are provided in section "Investigating Transfer Boundaries" of the Appendix.
>
> **Why magnitude of singular value is a proxy of transferability?**
>
> To answer this question, we propose two experiments. In the first one, we fine-tuned only one singular value from $\Delta_m$ at a time, starting from the one with the highest magnitude.The results averaged over all target tasks reveals a downward trend: the top-ranked component yields the highest performance with 61.64% accuracy, followed by a progressive decline for lower-ranked components dropping to approx. 30-40% (see Fig. 11 of the Appendix).
> The second experiment measures similarity between matrices generated by each component from $\Delta_m$ to the matrix obtained by fine-tuning the model to target task (upperbound, not available during training). As presented in Fig. 12 of the Appendix, matrices of top-ranked components are more similar to target matrix, which justifies using magnitude as a proxy of transferability. Details in “Transferability by SVD components” section of the Appendix.
>
> **Increasing clarity of Fig. 3 and adding table with detailed results**
>
> We have revised Fig. 3 to clearly indicate the superior performance of AXIS and we added its tabular correspondence in Table 4 of the Appendix. Moreover, we would like to remark that superior performance of AXIS for various datasets is also presented in Tables 6, 7, and 8 of the original Appendix.

---

### Author Response · Authors · 2025-11-28
**Official Comment by Authors**

We would like to thank all the Reviewers and the Chairs for their hard work. We have carefully read the reviews and addressed all the issues raised, which undoubtedly improved the quality of our submission.
We uploaded the revised version of our work and are ready to participate in further discussion. Based on comments, we have made the following changes to the revised version:
- We demonstrate the effectiveness of AXIS on language tasks (Fig. 10).
- We show that AXIS operates effectively in scenarios beyond same-initialization, including transfer across different architecture variations and scales (Table 6).
- We justify our rationale for selecting primary singular components (Fig. 4, Fig. 11, Fig. 12, Fig. 15).
- We reinforce the position of AXIS relative to other methods by demonstrating its robustness against realistically corrupted source models and partial information (Fig. 4).
- We highlight the superiority of optimizing singular values over optimizing other SVD elements (Table 5).
- We have significantly improved the presentation quality, clarity and readability of the paper.

We thank the Reviewers once again and look forward to discussing any other aspects of the paper that require further clarification.

--- authors

---

### Meta-Review · Area_Chair_FPvq · 2025-12-16

**Summary:**

Reviewer Xmc5 expressed concerns on the direction alignment between diverse sources and target task, thus limiting adaptation flexibility with only finetuning of singular vectors. Reviewer uXKj pointed out its limitation on heterogeneous architectures and missing experiments on non image classification tasks. Reviewer eMxV argued the soundness of motivation about cross-architecture model merging. Reviewer R9qd challenged the motivation and rationale of the proposed method, then pointed out a similar related work. Reviewer LXNa believed that the assumption that all source models share the same architecture and originate from the same pretrained weights is restrictive and unrealistic.

The AC agrees the concerns proposed by Reviewer Xmc5, Reviewer uXKj and Reviewer LXNa. The effectiveness of the proposed method heavily replies on the shared pretrained backbone. The weights that comes from different architectures or pretrained weights can not be well aligned for SVD decomposition of model merging. The Reviews converge to this issue. So the AC recommends it as rejection.

**Reviewer Concerns:**

The concerns about weight alignment of different architectures or pretrained weights are not well addressed.

**Reviewer Scores:**

Reviewer Xmc5: 4

Reviewer uXKj: 4

Reviewer eMxV: 4

Reviewer R9qd: 4

Reviewer LXNa: 4

---

### Decision · Program_Chairs · 2026-01-26

Reject